# Condensation tendency and planar isotropic actin gradient induce radial alignment in confined monolayers

**Tianfa Xie[1†], Sarah R St Pierre[1†], Nonthakorn Olaranont[2†], Lauren E Brown[3], Min Wu[2]\*, Yubing Sun[1,3,4]\***

[1]Department of Mechanical and Industrial Engineering, University of Massachusetts, Amherst, United States; [2]Department of Mathematical Sciences, Worcester Polytechnic Institute, Worcester, United States; [3]Department of Biomedical Engineering, University of Massachusetts, Amherst, United States; [4]Department of Chemical Engineering, University of Massachusetts, Amherst, United States

**Abstract** A monolayer of highly motile cells can establish long-range orientational order, which can be explained by hydrodynamic theory of active gels and fluids. However, it is less clear how cell shape changes and rearrangement are governed when the monolayer is in mechanical equilibrium states when cell motility diminishes. In this work, we report that rat embryonic fibroblasts (REF), when confined in circular mesoscale patterns on rigid substrates, can transition from the spindle shapes to more compact morphologies. Cells align radially only at the pattern boundary when they are in the mechanical equilibrium. This radial alignment disappears when cell contractility or cell-cell adhesion is reduced. Unlike monolayers of spindle-like cells such as NIH-3T3 fibroblasts with minimal intercellular interactions or epithelial cells like Madin-Darby canine kidney (MDCK) with strong cortical actin network, confined REF monolayers present an actin gradient with isotropic meshwork, suggesting the existence of a stiffness gradient. In addition, the REF cells tend to condense on soft substrates, a collective cell behavior we refer to as the 'condensation tendency'. This condensation tendency, together with geometrical confinement, induces tensile prestretch (i.e. an isotropic stretch that causes tissue to contract when released) to the confined monolayer. By developing a Voronoi-cell model, we demonstrate that the combined global tissue prestretch and cell stiffness differential between the inner and boundary cells can sufficiently define the cell radial alignment at the pattern boundary.

**\*For correspondence:**
mwu2@wpi.edu (MW);
ybsun@umass.edu (YS)

[†]These authors contributed equally to this work

**Competing interests:** The authors declare that no competing interests exist.

## Introduction

The collective migration and rearrangement of cells play a critical role in various biological processes such as morphogenesis (*Friedl et al., 2004*), wound healing (*Gurtner et al., 2008*), and cancer metastasis (*Trepat et al., 2012*). For epithelial cells which have well-defined cell-cell junctions and cortical actin network, biomechanics of the monolayer has been characterized in detail (*Trepat and Sahai, 2018*; *Tambe et al., 2011*; *Trepat and Fredberg, 2011*; *Rodríguez-Franco et al., 2017*). Further, polygonal-cell-based models, such as Vertex models and Voronoi models have been developed to understand how cell-cell adhesion, contractility, and self-propulsion determine cell topography (*Farhadifar et al., 2007*), single-cell motility (*Bi et al., 2016*), and collective migration (*Tetley et al., 2019*). Recently, theories of active nematic liquid crystals have been applied to cell monolayers to understand the cell organization in various cell types such as NIH-3T3 fibroblasts (*Duclos et al., 2017*), neural progenitor cells (*Kawaguchi et al., 2017*), and myoblast cells (*Blanch-Mercader et al., 2021a*; *Blanch-Mercader et al., 2021b*). A general observation obtained in these cell systems is that cells align their shapes with one another, and eventually form half-integer (*Duclos et al., 2017*;

*Kawaguchi et al., 2017*; *Saw et al., 2017*) and single-integer (*Blanch-Mercader et al., 2021a*; *Blanch-Mercader et al., 2021b*; *Guillamat et al., 2020*) topological defects. In these works, individual cells are treated as elongated and self-propelled particles, and the long-range alignment of these cells is well described by the hydrodynamic theory of active gels and fluids. It has been revealed that cells located at topological defects experience large compressive stress which leads to the apoptotic exclusion of cells, differentiation, and establishment of out-of-plane tissue architecture (*Saw et al., 2017*; *Guillamat et al., 2020*).

Distinct from spindle-like fibroblasts with minimal intercellular interactions or epithelial cells with cortically distributed actin cytoskeleton, various mesenchymal cells such as neural crest cells (*Achilleos and Trainor, 2012*), mesenchymal stem cells (*Aomatsu et al., 2014*), and chondrocytes (*Tavella et al., 1994*), express a significant level of intercellular adhesion molecules like N-cadherin and have isotropic actin network. It is unclear how the mechanical interactions among these cells contribute to their collective behaviors and biological functions.

In this work, we use a rat embryonic fibroblast (REF) cell line (REF-52) as a model system for the aforementioned mesenchymal cells to study their collective behaviors under mechanical equilibrium states when cell motility reduces significantly. Unlike 3T3 fibroblasts, REF cells express a significant level of cell adhesion molecules, such as N-cadherin and β-catenin (*Mary et al., 2002*), and have isotropic actin meshwork in contrast to cortically distributed actin in epithelial cells. We demonstrate that REF cells can robustly form radial alignment and show significant area expansion at pattern boundaries when confined in circular mesoscale patterns on a rigid substrate. When cultured on soft substrates, the REF monolayer retracts, and cells aggregate to the pattern centers, a collective cell behavior we term as 'condensation tendency' because this is a reminiscence of the mesenchymal condensation phenomenon, a prevalent morphogenetic transition in development that involves the aggregation of mesenchymal cells (*Mammoto et al., 2011*; *Klumpers et al., 2014*; *Lim et al., 2015*). Notably, such condensation tendency is similar to the recently discovered 'active dewetting' process in a monolayer of MDA-MB-231 cells with inducible E-cadherin, where cell contractility and intercellular adhesion drive the retraction of the tissue monolayer (*Pérez-González et al., 2019*). New to the active dewetting process, we show that when confined in circular mesoscale patterns on rigid substrates, the boundary REF cells can robustly develop radial alignment in mechanical equilibrium. By developing a Voronoi cell model (*Bi et al., 2016*; *Olaranont, 2019*), we discover that such condensation tendency, together with an autonomously established tissue-scale actin gradient, are sufficient to explain the observed radial alignment in confined circular geometries. The distinct behavior of REF-like cells may play a functional role in the collective migration and mesenchymal condensation, which are conventionally considered driven by chemotaxis (*Takebe et al., 2015*; *Szabó and Mayor, 2018*).

## Results

### REF 2c cells align radially on circular mesoscale patterns

We first sought to investigate whether various types of cells with different intra- and intercellular forces and cell shapes have similar self-organization behavior under confinement. We found that when REF 2c cells, a subclone of the REF52 cell line, were placed on circular micro-contact printed patterns (diameter = 344 μm), the boundary cells radially aligned over a period of 48 hr after cell seeding (*Figure 1A*). Video tracking of individual inner and boundary cells showed that the radial alignment of boundary cells was not caused by oriented cell migration toward or away from the center (*Figure 1—figure supplement 1A*, *Figure 1—video 1*), as most boundary cells stayed on the periphery of the pattern. Moreover, both the innermost and boundary cells migrated longer and faster between 24 and 36 hr than 36 and 48 hr (*Figure 1—figure supplement 1B–C*). This reduced cell migration coincided with the formation of radial alignment of cells on the pattern boundary. In contrast, 3T3 fibroblasts maintained circumferential alignment on the boundary at both 24 and 48 hr (*Figure 1A*), consistent with previous reports. To quantify the cell alignment, we traced the cell outlines and measured the cell angle deviation, defined as the angle between the long axis of an ellipse-fitted cell and the line that connects the pattern center and the centroid of the cell; cell elongation, defined as the ratio of the major axis to the minor axis of the ellipse-fitted cells; and projected cell area using an ellipse-fitting-based method (*Figure 1—figure supplement 2*). We found

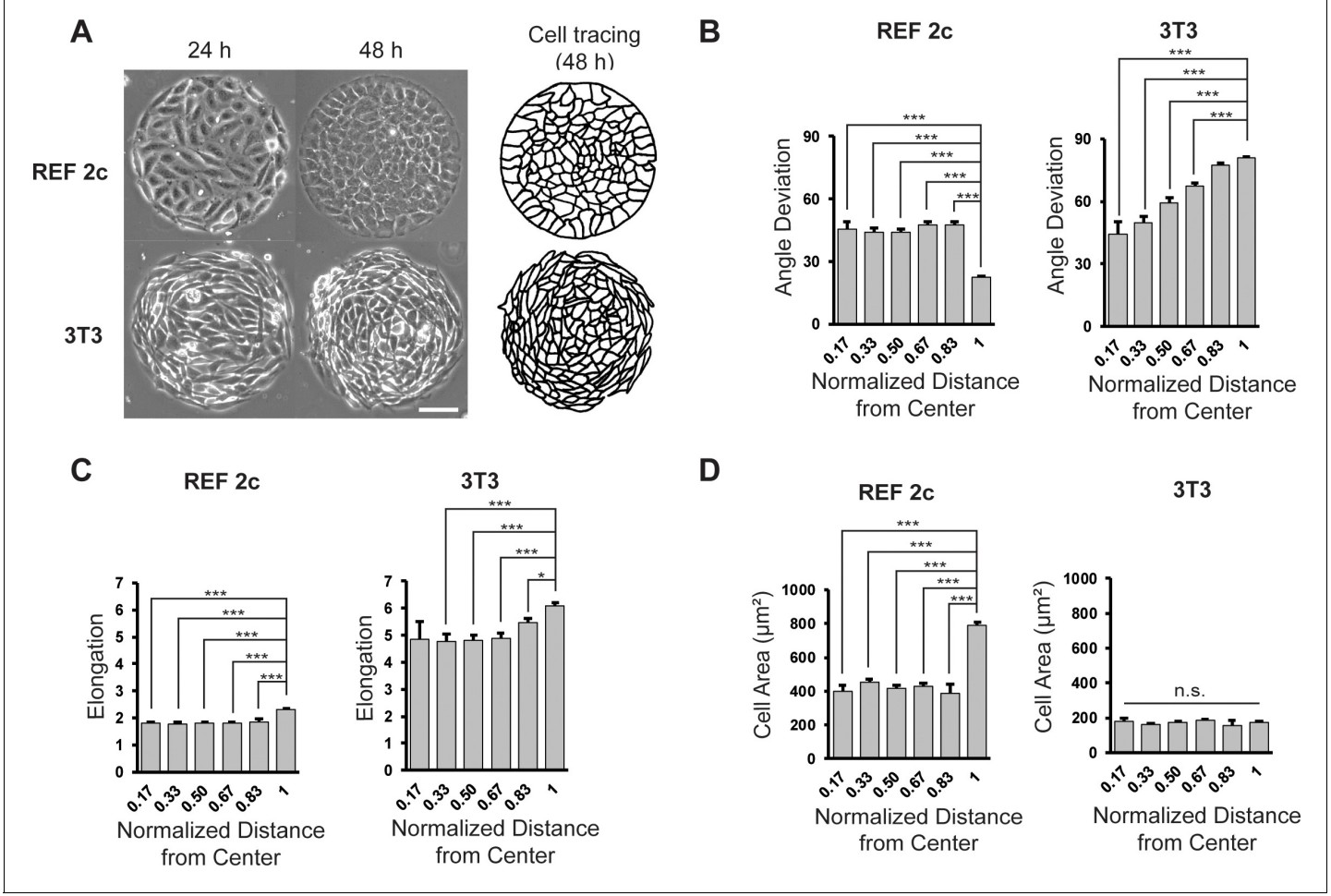

**Figure 1.** REF 2c cells align radially on circular mesoscale patterns. (**A**) Phase images showing REF 2c and 3T3 cells cultured on patterns for 24 and 48 hr. Scale bar: 100 μm. Cell tracing showed the cell boundaries at 48 hr. (**B**) REF 2c and 3T3 cell angle deviation at 48 hr as a function of normalized distance from the center of the pattern. $n$ = 16 patterns. (**C**) REF 2c and 3T3 cell elongation at 48 hr as a function of the normalized distance from the center of the pattern. $n$ = 16 patterns. (**D**) REF 2c and 3T3 cell area at 48 hr as a function of normalized distance from the center of the pattern. Cell area is defined as the area of the ellipse-fitted cells. $n$ = 16 patterns. Data are represented as mean ± s.e.m. *, p < 0.05; ***, p < 0.001; *n.s.*, p > 0.05. The online version of this article includes the following video and figure supplement(s) for figure 1:

**Figure supplement 1.** Migration of REF 2c cells drastically decreased 36 hr after cell seeding.

**Figure supplement 2.** Image analysis method for the quantification of angle deviation, cell elongation, and cell area.

**Figure supplement 3.** REF 2c cells became more compact at 48 hr compared with 24 hr.

**Figure supplement 4.** The thickness of micropatterned REF 2c colony increased from 24 hr to 48 hr.

**Figure supplement 5.** REF 2c cells condense on soft substrates.

**Figure 1—video 1.** A time-lapsed recording of the REF 2c cells migrating on a circular pattern from 24 hr to 48 hr after cell seeding.

https://elifesciences.org/articles/60381#fig1video1

that at 48 hr, the REF 2c cells were more compact compared to 24 hr, and much more compact than 3T3 cells, as indicated by the smaller elongation parameter (*Figure 1C*, *Figure 1—figure supplement 3*). Further, for REF 2c cells, the boundary cells were significantly more aligned with the radial direction than the inner cells at 48 hr, although these cells showed circumferential alignment similar to 3T3 cells at 24 hr (*Figure 1B*). Boundary cells were also significantly more elongated and had a significantly larger cell area than inner cells (*Figure 1C,D*). The boundary 3T3 cells were significantly more circumferentially aligned and elongated than the inner cells but did not have any significant differences in cell area (*Figure 1B–D*). Confocal fluorescence images of REF 2c cells stained with a cell membrane permeable dye CellTracker-Green suggested that the average tissue thickness increased slightly due to the cell radial alignment, while the boundary cells had a wedge-shape with a

drastically reduced cell thickness on the pattern boundary side (*Figure 1—figure supplement 4*). The change in tissue thickness suggests cell volume conservation as the whole pattern condensed slightly while boundary cell area increased.

As cells were patterned on PDMS substrates (Young's modulus $E$ = 2.5 MPa), which are significantly stiffer than physiological extracellular matrices, we next tested whether a similar phenomenon could be observed on substrates with physiologically relevant stiffness. Here we applied a well-established PDMS micropost array (PMA) system with identical surface geometry and different post heights to tune substrate rigidity (*Figure 1—figure supplement 5*; *Sun et al., 2014*). We found that on soft PMA substrates ($E$ = 5 kPa, post height = 8.4 μm), REF 2c cells became polarized at 48 hr and condensed towards the center of patterns, resulting in reduced total cell area on each pattern. On stiff PMA substrates ($E$ = 1 MPa, post height = 0.7 μm), however, the total cell area on each pattern did not change between 4 and 48 hr, and only cells on the boundary became radially aligned, which is consistent with the results on flat PDMS substrates. To accurately quantify the cell shape and angle deviation changes, we used rigid ($E$ > 1 MPa) PDMS substrates for the rest of the experiments.

## Cell contractility and cell-cell adhesion are required for radial alignment

As all previous works demonstrated circumferential alignment of non-epithelial cells on circular patterns, we asked what factors contribute to the radial alignment of REF 2c cells. We isolated and expanded several subclones of the REF cell line, and identified one subclone, named REF 11b, which did not radially align at 48 hr on circular patterns (*Figure 2A*). RNA-seq data revealed that while most genes have similar expression levels in REF 11b and REF 2c subclones, a small subset of genes were expressed significantly differently (*Figure 2—figure supplement 1*). As some of these genes are associated with cell-substrate adhesion and contraction (e.g. *INTEGRIN* α7 and α8, *MYL9*, and *COL16A1*), we then measured the cell contractility of these two subclones. Traction force measurements for single cells of REF 2c and 11b showed that for both total force and force per area, REF 11b cells were significantly less contractile than REF 2c cells (*Figure 2B*). To verify that the cell contractility was required for radial alignment, we treated patterned REF 2c cells with Blebbistatin (to inhibit myosin ATPase activity) and Y27632 (to inhibit Rho kinase activity), drugs known to reduce cell contractility (*Beningo et al., 2006*), at 24 hr, and then imaged the patterned cells at 48 hr (*Figure 2C*). The boundary cells of the Blebbistatin and Y27632-treated groups were not significantly more radially aligned than the inner cells, compared to the vehicle control group with DMSO (*Figure 2C*). These results suggest that cell contractility is required for radial alignment formation.

As 3T3 cells have a similar level of contractility compared with REF 2c cells (*Ghibaudo et al., 2008*), we rationalized that other factors must also contribute to the radial alignment. Staining with β-catenin suggested that there were cadherin-mediated cell-cell interactions among REF 2c and REF 11b cells while overlapping and empty spaces could be found among 3T3 cells without clear cell-cell junctions (*Figure 2D*, *Figure 2—figure supplement 2*). These observations suggested that such cell-cell adhesion may be required for the establishment of radial alignment. To confirm this, we treated REF 2c cells with EGTA, which reduced cadherin-based cell-cell adhesion without significantly affecting cell adhesion to substrates (*Pérez-González et al., 2019*; *Ohgushi et al., 2010*; *Rothen-Rutishauser et al., 2002*; *Al-Kilani et al., 2011*; *Koutsouki et al., 2005*; *Charrasse et al., 2002*), at 24 hr and then examined the cell alignment at 48 hr. We found that when treated with EGTA, the boundary cells were significantly more circumferentially aligned than the inner cells (*Figure 2E*). To confirm that EGTA treatment did not significantly affect cell substrates interactions, we measured the contractility of single cells using the PMAs. As shown in *Figure 2—figure supplement 3*, EGTA treatment did not change the total traction force per cell or per cell area, suggesting calmodulin-mediated contractility does not predominate this system. Together, we identify that cell contractility and cell-cell adhesion are two essential factors required for the establishment of radial alignment of patterned cells. Importantly, REF 2c cells form aggregates on soft substrates and show similar tendency on rigid substrates (reflected by the radial alignment). We define this collective cell behavior as 'condensation tendency', which requires both cell contractility and cell-cell adhesion.

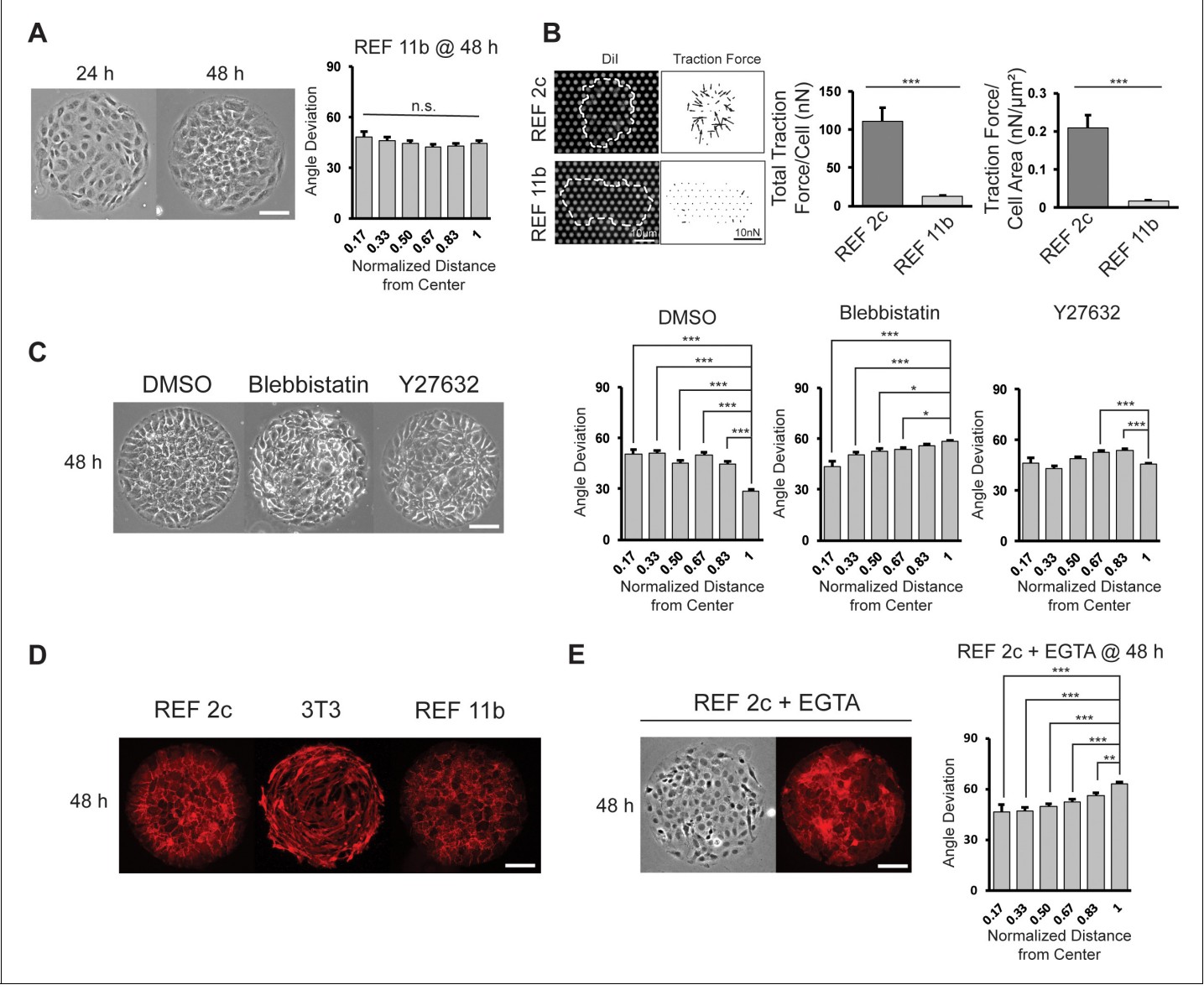

**Figure 2.** Effect of cell contractility and cell-cell adhesion on cell alignment. (**A**) Phase images of REF 11b at 24 hr and 48 hr after cell seeding. Average cell angle deviation is quantified with respect to distance from the center of the pattern. n = 16 patterns. Scale bar: 100 µm. (**B**) Representative images showing single REF cells cultured on PMA substrates and vector map of deduced traction forces. Plots show the total traction force per cell and traction force per cell area of individual REF 2c and 11b cells. n = 15 cells per subclone. (**C**) Phase images of REF 2c treated with DMSO, Blebbistatin, and Y27632 at 48 hr. Average cell angle deviation is quantified with respect to distance from the center of the pattern. n = 16 patterns per group. Scale bar: 100 µm. (**D**) Immunofluorescence images showing the expression of β-catenin in REF 2c, 3T3, and REF 11b cells at 48 hr. Scale bar: 100 µm. (**E**) Phase and fluorescence images showing cell orientation and β-catenin expression in EGTA treated REF 2c cells at 48 hr. Average cell angle deviation is quantified with respect to distance from the center of the pattern. Scale bar: 100 µm. Data are represented as mean ± s.e.m. *, p < 0.05; **, p < 0.01; ***, p < 0.001; n.s., p > 0.05.

The online version of this article includes the following figure supplement(s) for figure 2:

**Figure supplement 1.** Summary of gene expression differences in REF 11b compared to REF 2c.

**Figure supplement 2.** The cell morphology and distribution of β-catenin and actin are distinct between patterned 3T3 and REF 2c cells.

**Figure supplement 3.** EGTA treatment did not interfere cell-matrix interactions.

**Figure supplement 4.** Inhibiting cell proliferation does not change radial alignment of REF 2c cells.

## Cell proliferation is not required for the formation of cell radial alignment

It is notable that cell proliferation between 24 and 48 hr after cell seeding may lead to the remodeling and dissipation of elastic energy. To test this, we treated REF 2c cells with aphidicolin (1 μg/ml), an inhibitor of DNA replication (*Ikegami et al., 1978*). We found that with aphidicolin treatment, the cell number did not increase significantly from 24 to 48 hr (*Figure 2—figure supplement 4A,B*). In this condition, we still observed the radial alignment of boundary cells, which is quantitatively comparable with untreated controls (*Figure 2—figure supplement 4C*). Thus, we believe that the cell proliferation mediated remodeling is unimportant for the formation of cell radial alignment.

## A graded isotropic actin meshwork was established in patterned REF cells

The REF 2c tissue is reminiscent of the epithelial cell monolayers with coherent intercellular junctions. However, previous studies showed that no alignment was found when epithelial cells were confined on similar circular patterns (*Doxzen et al., 2013*). A critical difference between epithelial cells and fibroblasts is that the actomyosin network mainly distributes on the cell-cell boundaries in epithelial cell monolayer with apical-basal polarity while fibroblast cells have continuous actin meshwork. Thus, we investigate the spatial organization of actin filaments and the concentration of phosphorylated myosin light chain in confined REF monolayers. Surprisingly, we found a decrease in actin intensity occurred near the boundary of patterns for REF 2c cells, in the same location as the transition between isotropically oriented inner cells and radially aligned boundary cells. (*Figure 3*). To evaluate the statistical significance of this intensity gradient, we divided the pattern into six segments with the same width, and quantified actin intensity and actin intensity per cell as a function of distance to pattern center (*Figure 3—figure supplement 1A*). Our Analysis of Variance (ANOVA) results clearly demonstrated that the actin intensity in the outmost layer was significantly lower than that of the inner cells. To exclude the possibility that such actin intensity gradient is simply a result of cell density difference, we also quantified the total actin intensity and actin intensity per cell in inner cells and the outmost boundary cells. Consistently, a significant difference was found between inner and boundary cells, suggesting this actin intensity gradient was not simply a result of cell density differences (*Figure 3—figure supplement 1B,C*). In contrast, for REF 11b, there was a continued decrease in intensity from the center of the pattern to the outermost edge (*Figure 3*). This is in sharp contrast to the actin distribution of 3T3 cells, in which no actin gradient was identified (*Figure 3*). Notably, the sharp drop in the intensity profiles near the very end of the intensity plot for actin was an artifact as the outermost cells did not fully cover the pattern boundary. We further analyzed the correlation between actin gradient in individual patterns and corresponding angle deviation of boundary cells (*Figure 3—figure supplement 2*). Our results showed that increasing steepness of the actin gradient negatively correlates with boundary cell angle deviation ($r = - 0.533$), suggesting actin gradient may contribute to the boundary cell radial alignment.

## A Voronoi-cell model demonstrated that the condensation tendency with actin gradient is sufficient for establishing cell radial alignment at the pattern boundary

We next asked what are the deterministic factors that dictate the polar alignment of REF 2c cells. On one hand, the experimental data suggest that the condensation tendency is essential for the polar alignment (*Figure 2*). In addition, the data shows that the alignment is associated with the elongation of the boundary cells (*Figure 1*), which suggests that these boundary cells are stretched more along the radial direction than along the circumferential direction. On the other hand, the actin gradient decreasing towards the boundary (*Figure 3*) suggests that there is a stiffness differential between the inner and boundary cells, given it is well-documented that actin intensity is proportional to the local cell stiffness (*Tavares et al., 2017*; *Rotsch and Radmacher, 2000*; *Nawaz et al., 2012*; *Gavara and Chadwick, 2016*). To examine the possibility that increased actin intensity is associated with increased contractility, we examined the distribution of contractility levels in confined REF 2c cells by immunostaining phosphorylated myosin (p-myosin). It was found that unlike actin distributions, the p-myosin distribution was almost uniform across the whole pattern, with only a small increase near the pattern boundary (*Figure 3—figure supplement 3*). Thus, we hypothesize that the

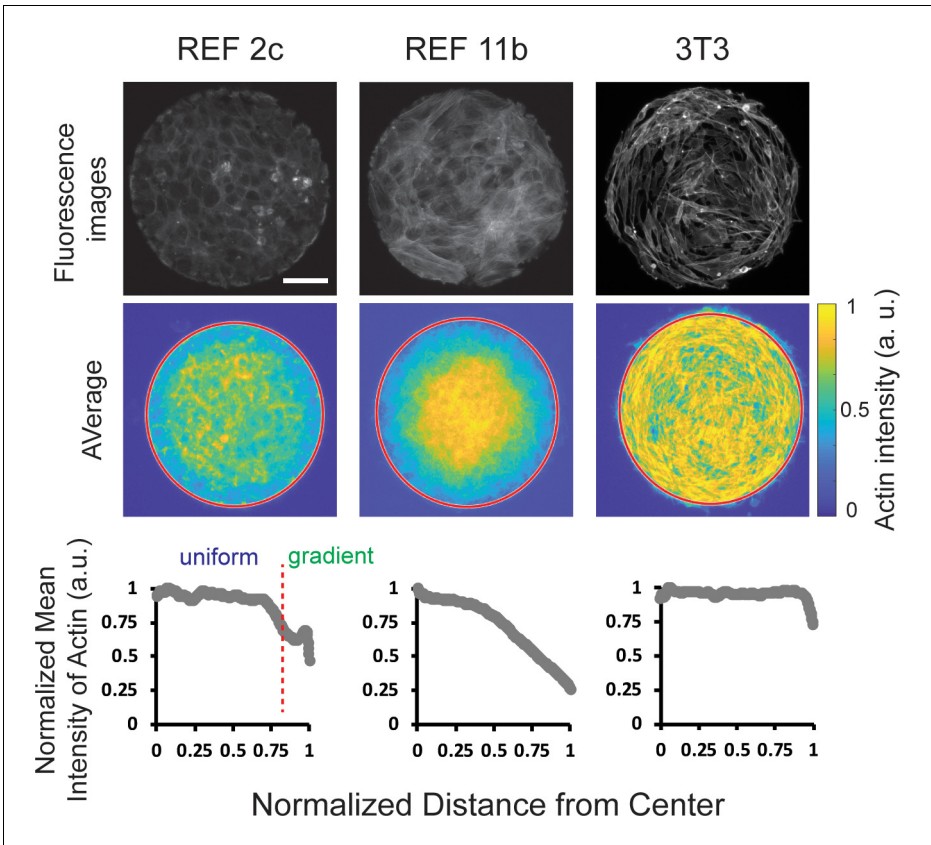

**Figure 3.** Tissue-scale actin intensity gradients were observed in patterned REF cells but not in 3T3 cells. Top: fluorescence images showing the actin staining of REF 2c, REF 11b, and 3T3 cells. Scale bar: 100 μm. Middle: colorimetric maps showing the average actin intensity profiles obtained by overlapping actin staining images. $n$ = 20 patterns per group. Bottom: normalized mean intensity of these overlapping images plotted as a function of the normalized distance from the center of the pattern.

The online version of this article includes the following figure supplement(s) for figure 3:

**Figure supplement 1.** Quantification of the actin intensity of patterned REF 2c cells confirmed the actin gradient.

**Figure supplement 2.** The correlation between actin gradient and radial alignment of REF 2c cells.

**Figure supplement 3.** Myosin-mediated total cell contractility slightly increased near pattern boundary.

**Figure supplement 4.** Isotropic actin network in REF 2c circular patterns.

radial alignment and elongation at the boundary in mechanical equilibrium is due to the condensation tendency and the cell stiffness differential between the inner and boundary cells.

We developed a Voronoi-cell model (*Bi et al., 2016*; *Olaranont, 2019*) to test our hypothesis (see Materials and methods). Given the actin orientation being neither radially nor circumferentially aligned (see *Figure 3—figure supplement 4* and Materials and methods), we assume that the stiffness of the cells is isotropic. Similarly, we introduce an isotropic prestretch parameter $0 < g \leq 1$ to describe the global condensation tendency in the patterned fibroblasts, which is a factor of the intrinsic cell area. The more $g$ deviates from 1, the larger the condensation tendency. As suggested by the direction of actin gradient (*Figure 3*), we introduce the stiffness differential between the boundary and the inner cells as $0 < \rho \leq 1$. Noticeably, although the mechanical interaction between the cell monolayer and the substrate has been experimentally measured previously (*Tambe et al., 2011*), it is challenging in our case because to accurately measure the traction forces using PMA substrates, the effective Young's modulus of the substrates needs to be less than 5 kPa, on which REF cells form aggregates due to the condensation tendency we described previously. Thus, we consider the mechanical equilibrium as a result of cell-cell mechanical interactions and the confinement at the tissue boundary (see Materials and methods for numerical implementation).

Our simulations show the radial alignment emerges when both condensation tendency $g$ and stiffness differential $\rho$ deviate from 1 (**Figure 4A**, **Figure 4—figure supplement 1**). Our shape analysis shows that in the radial-alignment case, the boundary cells are larger in size and more elongated (**Figure 4B**), consistent with the experiments (see **Figure 1C,D**, REF 2c cells). The results also explain the cell morphology observed in REF 11b cells, where no radial alignment is observed without contractility (see **Figure 4B** and **Figure 4—figure supplement 1A**, top rows). In addition, we show that when the value of $g$ is smaller in the boundary cells than that in the inner cells, and without stiffness differential ($\rho = 1$), boundary cells have smaller areas but do not align in the radial direction (**Figure 4—figure supplement 2**). This suggests that the slight elevation of p-myosin in the boundary cells (**Figure 3—figure supplement 3**) is not the cause of the radial alignment.

The stress and strain analysis on individual cells from simulations show that the boundary cells and the inner cells are under different mechanical cues (**Figure 4—figure supplement 3**). Compared to the inner cells, the boundary cells are under more significant and anisotropic strains but more minor tensile stresses, on average. The differences in mechanical cues, especially the anisotropic stretch of the boundary cells might induce the actin cytoskeleton reorganization in cells. Thus, the observed radial alignment may also be due to the adaption of cell shapes to mechanical cues. To evaluate this possibility, we stained a focal adhesion protein, vinculin (**Goldmann, 2016**; **Figure 4—figure supplement 4**), and a canonical mechanosensor, YAP (**Dupont et al., 2011**) at 48 hr when radial alignment is prominent (**Figure 4—figure supplement 5**). If mechanotransduction is involved,

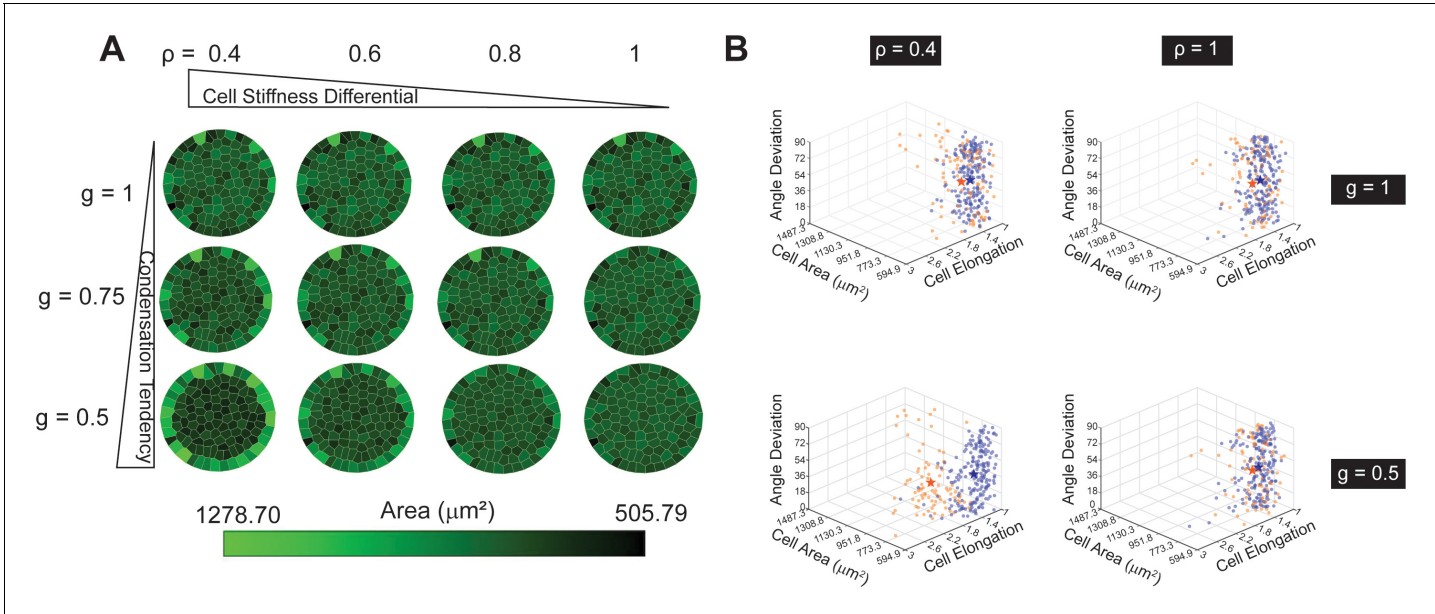

**Figure 4.** Condensation tendency and cell stiffness differential explain the radial-alignment boundary. (**A**) Voronoi cell modeling results from varying condensation tendency parameter $g$ and cell stiffness differential parameter $\rho$. Smaller $g$ and $\rho$ contribute to larger cell area along the boundary. (**B**) 3D Scatter plots of three cell-morphological parameters: angle deviation, cell area, and cell elongation. The three parameters on inner (orange) and boundary cells (blue) become significantly different when both $\rho$ and $g$ are small (e.g. $\rho = 0.4$ and $g = 0.5$; the stars represent the mean values among each cell group). $n$ = 5 patterns.

The online version of this article includes the following figure supplement(s) for figure 4:

**Figure supplement 1.** Quantitative analysis of the Voronoi cell modeling for circularly patterned cells.

**Figure supplement 2.** Increased boundary contractility is not essential to the radial-alignment boundary.

**Figure supplement 3.** Stress and strain analysis of the simulation from the Voronoi cell modeling for circularly patterned cells.

**Figure supplement 4.** Uniform vinculin expression in patterned REF 2c cells.

**Figure supplement 5.** YAP is activated in REF 2c cells.

**Figure supplement 6.** Actin distribution of REF 2c, REF 11b and 3T3 cells confined on ring-shaped patterns.

**Figure supplement 7.** Ring-shape micropatterns present circumferentially oriented actin network.

**Figure supplement 8.** Cell alignment on ring-shaped patterns.

**Figure supplement 9.** Cell area of REF 2c cells on ring patterns.

we expect to see higher intensities of vinculin (*Sigaut et al., 2018*) and/or nuclear YAP (*Elosegui-Artola et al., 2017*) in the boundary cells. However, we found that the distribution of vinculin was uniform across the pattern area, except for reduced vinculin intensity near the pattern boundary. The overall vinculin intensity per cell in the outmost layer of cells was still comparable with inner cells. We also found that most of the REF 2c cells expressed nuclear YAP, which is independent of confinement and cell alignment status (24 hr vs. 48 hr). These results suggest local activation of these mechanotransduction pathways in boundary cells is unlikely the cause for cell radial alignment, although we cannot rule out the involvement of other mechanotransduction pathways, such as those mediated by mechanosensitive ion channels and certain G protein coupled-receptors (*Martino et al., 2018*; *Holle and Engler, 2011*; *Marullo et al., 2020*).

In summary, we have shown that condensation tendency with stiffness differential near the pattern boundary is sufficient to replicate the radial alignment and increased elongation and area of the boundary cells compared to inner cells in the REF 2c in vitro.

## Actin gradient and condensation tendency maintain under the change of tissue topology

Our results demonstrated that the emergence of a cell stiffness gradient along the radial direction is critical for cell alignment. We next investigated whether such gradient maintains at the outer boundary under the change of the topology. To do so, we designed two ring patterns with different inner diameters (200 μm and 300 μm) and the same outer diameter (400 μm). We found that surprisingly, both REF 2c and 11b cultured on ring patterns robustly showed an actin intensity gradient from the center to the boundary, regardless of the change of topology and the radius of the inner boundary. In contrast, 3T3 cells did not have an actin gradient for any ring patterns (*Figure 4—figure supplement 6*). Similar to the circular patterns, we calculated the actin fiber angle deviation and the structure parameter $k_H$ for REF 2c cells (*Figure 4—figure supplement 7*). Compared to the full circle pattern, the distribution of the structure parameter $k_H$ in ring patterns reveals that the actin fibers are mostly aligned along the tangential direction at the inner boundary and the tangential alignment decreases along the radius, suggesting a complex interaction between the actin network and the inner boundary. For the thicker ring pattern, the actin network almost becomes isotropic ($k_H \sim 0.65$) at the outer boundary. We then sought to investigate whether changing tissue topology to ring shapes change the REF 2c cell alignment. We found that for both thick and thin ring patterns, the outermost boundary REF 2c cells radially aligned, but not REF 11b cells or 3T3 cells, which are quantified using angle deviation (*Figure 4—figure supplement 8*). The cell area increased significantly near the exterior pattern boundary for REF 2c (*Figure 4—figure supplement 9*). Interestingly, the innermost boundary cells aligned circumferentially along the inner boundary.

## Discussion

There is a growing interest in understanding the physical principles of the autonomous collective behavior of cells. Epithelial cells are often modeled as polygon arrays, and the mechanical states are described by the active stress between neighboring cells (*Farhadifar et al., 2007*; *Bi et al., 2016*; *Tetley et al., 2019*). On the other hand, actin-rich mesenchymal cells are often modeled as self-propelled particles, and they are believed to self-assemble as active nematics (*Duclos et al., 2014*; *Doostmohammadi et al., 2018*). Our results revealed a new class of behavior of the actin-rich cells with significant intercellular adhesion, represented by REF cells. We showed that these cells could form compact morphology and displays a condensation tendency, resulting in a change of cell size and cell elongation at the tissue boundary on rigid substrates. Unlike skeletal muscle cells, such condensation tendency is not originated from the anisotropy of actin bundles, as the actin network in REF monolayer is found to be isotropic (*Figure 3—figure supplement 4*). Our theoretical analysis suggests that such condensation tendency of the tissue, together with an autonomously established stiffness gradient in the monolayer, are sufficient to form the radial cell alignment tissue boundary in the confined planar geometries. Disrupting the cell-cell adhesion using EGTA also abolished the radial cell alignment of REF 2c cells (*Figure 2E*).

In this work, we describe the cell monolayer mechanics associated with cell contractility and differential cell stiffness by developing a cell-based model. Vertex models (*Farhadifar et al., 2007*; *Tetley et al., 2019*) and Voronoi cell models *Bi et al., 2016*; *Olaranont, 2019* have been well-

established for epithelial-like cells with strong cell-cell adhesion (*Farhadifar et al., 2007*), because of the cortical distribution of actin cytoskeleton. In previous cell-based modeling, vertex models (*Farhadifar et al., 2007*; *Tetley et al., 2019*) and Voronoi cell models (*Bi et al., 2016*; *Olaranont, 2019*) often describe the epithelial cell contractility along the intercellular junctions due to the cortical distribution of actin cytoskeleton with strong cell-cell adhesion. In our case of REF monolayer where the actin network does not spread coincidentally with the intercellular junctions, we modify the Voronoi-cell model to describe the condensation tendency as a prestretch on individual cell area, which has not been considered in the previous vertex or Voronoi cell models. We have shown that condensation tendency and a stiffness differential between the inner and boundary cells are sufficient to establish the radial-cell-alignment tissue boundary.

The cell system we described here is different from epithelial cells, which also have strong intercellular adhesion and contractility as individual cells. Epithelial cells, when confined in circular patterns, form nematic symmetry that is similar to 3T3 cells (*Saw et al., 2017*; *Doxzen et al., 2013*). However, when they are confined in ring-shape patterns, chiral spiral defects can be found (*Wan et al., 2011*). The different cell alignment between epithelial cells and REF cells is likely due to the distribution of actin network. It is well-known that actin is mainly distributed in the cell-cell junctions for epithelial cells (*Fanning et al., 2012*), while actin stress fibers can be found in REF cells even when they form a compact monolayer. Notably, neuroepithelial cells derived from pluripotent stem cells could form rosettes-like structures in vitro, with tight and adherence junctions presented at the side facing the internal lumen (*Knight et al., 2018*; *Elkabetz et al., 2008*). However, the formation of neural rosettes may require a completely different mechanism as neuroepithelial cells are planar polarized (*Davey et al., 2016*), and the junctional proteins distributed homogenously for REF cells. Recently, Roux, Kruse, and colleagues reported that C2C12 myoblasts could self-organize into spiral patterns when confined using similar circular confinement (*Blanch-Mercader et al., 2021a*; *Blanch-Mercader et al., 2021b*; *Guillamat et al., 2020*). When cell proliferation is allowed, the system continues to evolve to aster pattern with boundary cells elongated and radially aligned (*Blanch-Mercader et al., 2021b*) and eventually out-of-plane mounds in pattern center (*Guillamat et al., 2020*). Using hydrodynamic descriptions, they attributed the formation of such patterns to both directional traction forces and nematic active stresses (*Blanch-Mercader et al., 2021a*; *Blanch-Mercader et al., 2021b*). Computational and experimental works also confirmed that the tissue is contractile and generates compressive forces to cells in the pattern center. While our observation of radial cell alignment on pattern boundary is reminiscent of the aster patterns observed in those works, an important difference is that the cell proliferation is not required for the radial alignment to appear (*Figure 2—figure supplement 4*). Further, the radial alignment in C2C12 cells seems to be an intermediate state between the spiral tissue flow to mound growth, while the REF radial alignment is an equilibrium state where the cell velocity is nearly zero (*Figure 1—figure supplement 1*). While cell proliferation is a critical driving factor in aster pattern formation in the C2C12 system, in our REF system, the stiffness differential is indispensable for the radial alignment formation. Together, the REF cells represented a unique class of cell systems that are different from typical spindle-like 3T3 cells, epithelial cells, and C2C12 myoblasts. Future studies will be needed to investigate whether other cells that may fall into this category, such as neural crest cells, chondrocytes, and epithelial cells undergoing epithelial-mesenchymal transition, also behave similarly on patterned substrates. Interestingly, neural crest cells migrate as a cohesive group mediated by self-secreted C3a (*Carmona-Fontaine et al., 2011*), and chondrocytes form mesenchymal condensation mediated by EGF signals (*Qin and Beier, 2019*). The condensation tendency of these cells may be relevant to their biological functions by facilitating their orientation towards the signal center.

In multiple developmental and physiological processes, previous focuses have been on the differential growth and contractility as a stress-and-strain driver (*Ambrosi et al., 2019*; *Streichan et al., 2018*), while there is a lack of elucidation of the role on differential stiffness (except in *Drosophila* gastulation [*Polyakov et al., 2014*; *Rauzi et al., 2015*]). In 2D tissues, it is difficult to disentangle differential contractility and stiffness as the two mechanical properties are both connected to the actomyosin activities. A positive correlation between the actin and myosin spatial distributions has been widely reported previously (*Martin et al., 2009*; *Banerjee et al., 2017*). However, in the REF microtissue, we have found that the myosin gradient is not evident and does not coincide with that of the actin gradient. While it is well-documented that myosin is associated with higher contractile force (*Murrell et al., 2015*) and actin is associated with higher stiffness (*Tavares et al., 2017*; *Rotsch and*

*Radmacher, 2000*; *Nawaz et al., 2012*; *Gavara and Chadwick, 2016*), in theory, we disentangled their roles in regulating the cell morphology at the REF tissue boundary. We reason that differential stiffness with uniform contractility is sufficient to explain the radial-cell-alignment tissue boundary. Indeed, the actin mesh architecture may also affect the contractile machinery, such as through the concentration of the crosslinking proteins (*Kasza and Zallen, 2011*). Thus, we cannot exclude the possibility of a contractility differential associated with the observed actomyosin intensity field. Notably, in a very recent work, the authors associated actin intensity differential with contractility differential during mesoderm invagination in the early *Drosophila melanogaster* embryo (*Denk-Lobnig et al., 2021*). However, this possibility requires further analysis on how myosin motors and actin network molecules interact to generate macroscopic network contraction. Future work will investigate how tissue-level actin gradient and orientation field and tissue flow interact in response to geometric and mechanical cues. In particular, a Maxwell-type viscoelastic continuum model will be needed to couple the actin dynamics with active tissue mechanics to explain the onset of tissue flow and the maintenance of the tissue mechanical equilibrium (*Wei and Wu, 2021*). This new framework will account for the energy dissipation due to cell-cell junction and cytoskeletal rearrangement during the tissue flow before equilibrium.

In summary, this work reports a unique behavior of REF cells that develop radial-alignment boundary when confined in circular and ring mesoscale patterns. The formation of such radial alignment is not a result of directed cell migration and requires a condensation tendency and an emergent cell stiffness differential. Future work must be done to understand the molecular mechanisms of this novel collective cell behavior and its contribution to relevant biological functions.

# Materials and methods

## Cell culture

Original Rat Embryo Fibroblast cell line (REF-52; RRID:CVCL_6848) stably expressing yellow fluorescent protein (YFP) - paxillin fusion protein is a gift from Dr. Jianping Fu. REF 11b and 2c subclones were generated by single-cell clone selection. Cells were maintained in high-glucose Dulbecco's modified Eagle's medium (DMEM, no glutamine; Invitrogen) supplemented with 10% fetal bovine serum (FBS; Invitrogen), 4 mM L-glutamine (Invitrogen), 100 units/mL penicillin (Invitrogen), and 100 µg/mL streptomycin (Invitrogen). NIH/3T3 cells (a gift from Dr. Mingxu You; RRID:CVCL_0594) were cultured in DMEM medium (with glutamine, Invitrogen) supplemented with 10% calf serum as suggested by ATCC, 100 units/mL penicillin (Invitrogen), and 100 µg/mL streptomycin (Invitrogen). Although not recommended, culturing 3T3 cells using media containing FBS did not significantly change experimental outcomes. Both 3T3 and REF cells were subcultured at about 80% confluency following standard cell culture procedures. All cells were cultured at 37°C and 5% $CO_2$. NIH-3T3 cells were authenticated using the STR profiling service provided by ATCC. All the cell lines have been tested for mycoplasma contamination using PCR-based methods and negative testing results were obtained.

## RNA sequencing and data analysis

Total RNA was extracted from REF subclones 2c and 11b using the Aurum Total RNA Mini Kit (Biorad) following the manufacturer's instructions. RNA quality was assessed using 6000 Nano Agilent 2100 Bioanalyzer (Agilent Technologies, Santa Clara, CA). The concentration of the libraries was measured using Qubit 3.0 fluorometer (Life Technologies, Carlsbad, CA). cDNA libraries were single-end sequenced in 76 cycles using a NextSeq 500 Kit v2 (FC-404–2005, Illumina, San-Diego, CA). High-throughput sequencing was performed using NextSeq500 sequencing system (Illumina, San-Diego, CA) in the Genomic Resource Laboratory of the University of Massachusetts, Amherst. All sequencing data were uploaded to the GEO public repository (https://www.ncbi.nlm.nih.gov/geo/) and were assigned series GSE148155. Validation of sequence quality was performed using the BaseSpace cloud computing service supported by Illumina (BaseSpace Sequence Hub, https://basespace.illumina.com/home/index). RNA-seq reads were aligned to the rat reference genome (Rattus norvegicus UCSC rn5) using TopHat Alignment. Then, the differential gene expression analyses were performed by Cufflinks Assembly and DE using previous alignment results produced by the TopHat app

as input. Shortlists of significantly differentially expressed genes were identified by applying thresholds of 2-fold differential expression and false discovery rate q $\leq$ 0.05.

## Cell migration assay

To track the migration of patterned REF 2c cells, brightfield live-cell imaging was performed at 37°C, 5% $CO_2$ using an automated digital microscope with a $10\times$ objective with a gas controller (Cytation three microplate reader, BioTek Instruments Inc, Winooski, VT, USA). Images were collected every 10 min for 24 hr. Acquired brightfield images were merged and corrected for frame drift. To analyze cell migration, individual cell positions were manually tracked using CellTracker software (*Piccinini et al., 2016*) implemented in MATLAB (MATLAB R2020a, MathWorks). The average speed for each cell was calculated as the total migration length of each cell divided by the total time. Mann–Whitney test was used to compare the migration length and average speed of the cells since the data were not normally distributed (Shapiro-Wilk test). Statistical differences were defined as where *, $p < 0.05$; **, $p < 0.01$; ***, $p < 0.001$.

## Microcontact printing

Soft lithography was used to generate patterned polydimethylsiloxane (PDMS) stamps from negative SU8 molds that were fabricated using photolithography. These PDMS stamps were used to generate patterned cell colonies using microcontact printing, as described previously (*Zhu et al., 2019*). Briefly, to generate patterned cell colonies on flat PDMS surfaces, round glass coverslips (diameter = 25 mm, Fisher Scientific) were spin-coated (Spin Coater; Laurell Technologies) with a thin layer of PDMS prepolymer comprising of PDMS base monomer and curing agent (10:1 *w/w*; Sylgard 184, Dow-Corning). PDMS coating layer was then thermally cured at 110°C for at least 24 hr. In parallel, PDMS stamps were incubated with a fibronectin solution (50 $\mu$g·ml$^{-1}$, in deionized water) for 1 hr at room temperature before being blown dry with a stream of nitrogen. Excess fibronectin was then washed away by distilled water and the stamps were dried under nitrogen. Fibronectin-coated PDMS stamps were then placed on top of ultraviolet ozone-treated PDMS (7 min, UV-ozone cleaner; Jetlight) on coverslips with a conformal contact. The stamps were pressed gently to facilitate the transfer of fibronectin to PDMS-coated coverslips. After removing stamps, coverslips were disinfected by submerging in 70% ethanol. Protein adsorption to PDMS surfaces without printed fibronectin was prevented by incubating coverslips in 0.2% Pluronic F127 solution (P2443-250G, Sigma) for 30 min at room temperature. Coverslips were rinsed with PBS before placed into tissue culture plates for cell seeding. For patterned cell colonies, PDMS stamps containing circular patterns with a diameter of 344 $\mu$m, and ring patterns with an outer diameter of 400 $\mu$m and inner diameter of either 200 $\mu$m (thick ring) or 300 $\mu$m (thin ring) were used.

## Immunocytochemistry

Four percent paraformaldehyde (Electron Microscopy Sciences) was used for cell fixation before permeabilization with 0.1% Triton X-100 (Fisher Scientific). Cells were blocked in 10% donkey serum for 1 hr at room temperature. Primary antibodies used were anti-β-catenin from rabbit (51067–2-AP, Proteintech), anti-α-tubulin from mouse (66031–1-Ig, Proteintech), anti-p-myosin from rabbit (3671T, Cell Signalling), anti-vinculin from mouse (V9131, Sigma), and anti-YAP from mouse (sc-101199, Santa Cruz). For immunolabeling, donkey-anti goat Alexa Fluor 488, donkey-anti rabbit Alexa Fluor 555, and donkey-anti mouse Alexa Fluor 647 were used. For actin filaments visualization, Alexa Fluor 488 conjugated phalloidin (Invitrogen) was used. Samples were counterstained with 4,6-diamidino-2-phenylindole (DAPI; Invitrogen) to visualize the cell nucleus.

## Small-molecule drugs treatment assays

Blebbistatin (10 $\mu$M, Cayman Chemical, cat.no.13013), Y27632 (10 $\mu$M, Cayman Chemical, cat. no.10005583), and Aphidicolin (1 $\mu$g/ml, Cayman Chemical, cat.no.14007) were dissolved in DMSO. Ethylene Glycol Tetraacetic Acid, (EGTA, 2 mM, Santa Cruz Biotechnology, cat.no. sc-3593A) was prepared in distilled water. Cells were treated with these drugs for 24 hr at 37°C.

## Traction force measurement

The protocol for generating microposts and measuring traction force has been published previously (*Xie et al., 2017*). For single cell traction force measurement, REF cells were incubated for 48 hr on DiI stained micropost arrays, then live-cell imaged. Microposts with a diameter of 2 μm and a height of 8.4 μm (effective modulus $E_{eff} = 5$ kPa) were used. Custom MATLAB script was written to quantify post deflection using *Equation 1*:

$$F = \left(\frac{3EI}{L^3}\right)x,$$

(1)

where $F$ is the force applied to the tip of the post, $E$ is the elastic modulus of PDMS, $I$ is the area moment of inertia, $L$ is the post height, and $x$ is the deflection of the post tip (MathWorks; https://www.mathworks.com/).

## Image analysis

Phase contrast and fluorescence images of patterned cell colonies were recorded using an inverted epifluorescence microscope (Leica DMi8; Leica Microsystems) equipped with a monochrome charge-coupled device (CCD) camera. Since the cell colonies were circle-like in shape and the approximate radii of the circles were known, the centers of the colonies could be found using the circle Hough transformation which is the MATLAB function *imfindcircles*. The distance vector between the center of each cell colony and the center of the image frame shifted each pixel of the image. Using the shifted images, the stacked images could be generated by adding the values at the same position of the pixels. The fluorescence intensity of each pixel in stacked images was normalized by the maximum intensity identified in each image. To plot average intensity as a function of distance from the pattern centroid, the stacked intensity maps were divided into 310, 186, and 91 concentric zones for the circle, thick ring, and thin ring, respectively, with single pixel width. The average pixel intensity in each concentric zone was calculated and plotted against the normalized distance of the concentric zone from the pattern centroid.

## Fiber angle deviation

The vector module of the Orientation J plug-in (*Rezakhaniha et al., 2012*) was used with Fiji (*Schneider et al., 2012*) to quantify the angle deviation of actin and α-tubulin stained fibers in REF cell images. Five images were analyzed for each group (circle, thick ring, thin ring). The mean fiber angle deviation was plotted versus the normalized distance from the center of the pattern. A step size of $172/31 \approx 5.55$ μm for the circle, $100/18 \approx 5.56$ μm for the thick ring, and $50/9 \approx 5.56$ μm for the thin ring patterns were used to generate approximately the same number of points respective to the size of the pattern. The schematic of (α), fiber angle deviation from the radius, was drawn in *Figure 3—figure supplement 4*.

## Structure tensor

The structure parameter, $k_H$, was calculated from histograms developed from the variation of fiber angle deviation at each concentric distance from the center or innermost edge of the patterns (*Figure 3—figure supplement 4* and *Figure 4—figure supplement 7*). We used a 2D structure tensor **H** to quantify the averaged fiber orientation of the fibers at each point along the radius (*Wu and Ben Amar, 2015*):

$$\mathbf{H} = \frac{1}{\pi}\int_{-\pi/2}^{\pi/2}\rho(\alpha)\mathbf{E}_\alpha \otimes \mathbf{E}_\alpha d\alpha = k_H\mathbf{I} + (1 - 2k_H)\mathbf{E}_R \otimes \mathbf{E}_R$$

(2)

where $\alpha$ is the angle between the outward radial direction $\mathbf{E}_R$ and actin fiber direction $\mathbf{E}_\alpha$ (See *Figure 3—figure supplement 4*) and $\rho(\alpha)$ is the normalized orientation density function satisfying $\rho(\alpha) = \rho(-\alpha)$ and $\frac{1}{\pi}\int_{-\pi/2}^{\pi/2}\rho(\alpha)d\alpha = 1$. As **I** is the 2 × 2 identity matrix and $\mathbf{E}_R \otimes \mathbf{E}_R = \begin{bmatrix} 1 & 0 \\ 0 & 0 \end{bmatrix}$, the structure tensor H can be represented by the structure parameter

$$k_H = \frac{1}{\pi} \int_{-\pi/2}^{\pi/2} \rho(\alpha)(\sin\alpha)^2 d\alpha. \qquad (3)$$

By definition, $0 \le k_H \le 1$, and the fiber distribution is more aligned with the radial (circumferential) direction as $k_H$ decreases (increases). $k_H = 0.5$ indicates that the actin fiber distribution is not aligned with either direction. From the actin intensity field, our quantification shows that $k_H$ is approximately 0.4 ~ 0.6 along the radius, suggesting an isotropic distribution of the fiber orientation (*Figure 3—figure supplement 4D*). The scripts for structure tensor calculation are available through Github, copy archived at swh:1:rev:0a5972451e8e747ea755cba6613ef0d81c8aabfd (*St Pierre and Wu, 2021*).

## Voronoi cell mathematical model

### Cell-tissue configurations

We begin with a 2D domain and generate polygonal cells from the 2D Voronoi tessellation to represent the cell configurations. The graph of Voronoi cell tessellation is dual to the Delaunay triangulations. In particular, the relation between a trio of adjacent Voronoi cell centers (<$r_i$, $r_j$, $r_k$>) and its corresponding vertex ($\omega_{<i,j,k>}$) is given by the following equations (*Bi et al., 2016*, *Olaranont, 2019*):

$$\vec{\omega}_{<i,j,k>} = a\vec{r}_i + b\vec{r}_j + c\vec{r}_k \qquad (4)$$

where

$$a = \left\| \vec{r}_j - \vec{r}_k \right\|^2 (\vec{r}_i - \vec{r}_j) \cdot (\vec{r}_i - \vec{r}_k)/D,$$

$$b = \left\| \vec{r}_i - \vec{r}_k \right\|^2 (\vec{r}_j - \vec{r}_i) \cdot (\vec{r}_j - \vec{r}_k)/D,$$

$$c = \left\| \vec{r}_i - \vec{r}_j \right\|^2 (\vec{r}_k - \vec{r}_i) \cdot (\vec{r}_k - \vec{r}_j)/D,$$

$$D = 2\left\| (\vec{r}_i - \vec{r}_j) \times (\vec{r}_j - \vec{r}_k) \right\|^2. \qquad (5)$$

## Cell prestretch and differential stiffness

To describe the monolayer mechanics, we define a total energy that is generally a function of the areas of cells $A^\alpha$ and lengths of junctions $l_\beta^\alpha$:

$$E_{total} = \sum_\alpha E^\alpha; E^\alpha = \frac{1}{2} K^\alpha \left( g A_0^\alpha \right) \left( \frac{A^\alpha}{g A_0^\alpha} - 1 \right)^2 + \sum_{\beta \in \Gamma} \lambda l_\beta^\alpha, \qquad (6)$$

where $\alpha$ indicates each cell, $\beta$ indexes junctions of cell $\alpha$, and $\Gamma$ is the set of junctions of cell $\alpha$ with tension. For modeling epithelial cells, the tension is considered on each intercellular junction to represent the net mechanical effect of actomyosin contraction and intercellular adhesion at the apical surface of the tissue. We apply tension to the intercellular junctions, and on the junctions located at the boundaries of the micropattern to ensure the circularity of the microtissue (see *Geometric confinement* below). To model the condensation tendency, we introduce the prestretch $0 < g \le 1$ which decreases the intrinsic cell size $A_0^\alpha$ of each cell. To describe the stiffness gradient between the boundary cells and interior cells (*Figure 4A*, the stiffness of boundary cells is smaller than or equal to that of the interior cells), we introduce the stiffness differential parameter $0 < \rho \le 1$, which is the ratio of the stiffness $K^\alpha$ between boundary and interior cells. To describe the contractility gradient between the boundary cells and interior cells (*Figure 4—figure supplement 2*), the contractility of boundary cells is larger than or equal to that of the interior cells, meaning the prestretch of the boundary cells is smaller than or equal to that of the interior cells, we introduce the contractility differential parameter $0 < \rho_g \le 1$, which is the ratio of the prestretch between boundary and interior cells.

## Mechanical equilibrium among cells

$E_{total}$ is a function of the coordinates of vertices via its dependence on lengths of junctions and areas of cells. In particular, the length of each junction is $l_\beta^\alpha = \|\vec{\omega}_{\beta,1} - \vec{\omega}_{\beta,2}\|$ where 1 and 2 indicate the adjacent vertices of $\beta$ junction, and the area of each cell is $A^\alpha = \frac{1}{2}\sum_{m=0}^{z^\alpha - 1}\|\vec{\omega}_m^\alpha \times \vec{\omega}_{m+1}^\alpha\|$, where $z^\alpha$ is the number of vertices of cell $\alpha$ (Notice that $\vec{\omega}_{z^\alpha}^\alpha = \vec{\omega}_0^\alpha$). In addition, the coordinates of vertices depend on the coordinates of a trio of neighboring Voronoi cell centers through *Equations (4) and (5)*. By the chain rule, we can solve the cell configurations by minimizing the energy $E_{total}$ following the dynamics of cell centers:

$$\frac{dr_x^\alpha}{dt} = F_x^\alpha \equiv -\frac{\partial E_{total}}{\partial r_x^\alpha} = -\left(\sum_m \frac{\partial E^\alpha}{\partial \omega_{mx}^a}\frac{\partial \omega_{mx}^a}{\partial r_x^\alpha} + \sum_m \frac{\partial E^\alpha}{\partial \omega_{my}^a}\frac{\partial \omega_{my}^a}{\partial r_x^\alpha}\right),$$

$$\frac{dr_y^\alpha}{dt} = F_y^\alpha \equiv -\frac{\partial E_{total}}{\partial r_y^\alpha} = -\left(\sum_m \frac{\partial E^\alpha}{\partial \omega_{mx}^a}\frac{\partial \omega_{mx}^a}{\partial r_y^a} + \sum_m \frac{\partial E^\alpha}{\partial \omega_{my}^a}\frac{\partial \omega_{my}^a}{\partial r_y^a}\right) \quad (7)$$

where $m$ is the index of the vertices of cell $\alpha$. The subscripts x and y indicate the component x or y of the vector we are considering.

## Geometric confinement at the boundary

We define a full circle or annulus inside the squared domain to initialize the micropattern, by calibrating the size ratio between individual cell and the micropattern according to the in vitro setup. Initially, there are 625 cells in the squared domain. From the experiments, the number of cells in the circular pattern is approximately 121 cells. Therefore, to simulate the results, the circular domain is demarcated by thresholding the distance of cells from the origin which covers around 120–124 cells depending on the random initial configuration. To round up the circular border, we minimize the following energy:

$$E_{initial} = \sum_\alpha E^\alpha + \frac{1}{2}k_b A_0 \left(\frac{A}{A_0} - 1\right)^2;$$

$$E^\alpha = \frac{1}{2}K^\alpha A_0^\alpha \left(\frac{A^\alpha}{A_0^\alpha} - 1\right)^2 + \sum_{\beta \in \Gamma}\lambda l_\beta^\alpha + \sum_{\beta \in \Gamma_{out}}\lambda_{out} l_\beta^\alpha$$

with $K^\alpha = 1, \lambda = 15, \lambda_{out} = 20$, and $k_b = 0.1$. The tension term $\sum_{\beta \in \Gamma_{out}}\lambda_{out} l_\beta^\alpha$ is included to ensure the circularity of the boundary, and the term $\frac{1}{2}k_b A_0 \left(\frac{A}{A_0} - 1\right)^2$ is included to ensure that the total area $A$ of the micropattern is close to the initial tissue area $A_0$. Once $E_{initial}$ reached a local minimum, the centroids of the boundary cells and cells outside of the circular domain were fixed. When the prestretch, stiffness differential and tension on the intercellular junctions are considered in *Equation (6)*, all the non-boundary cell centers move and reach equilibrium following *Equation (7)*.

## Cell geometry analysis

For the cell elongation parameter and angle deviation (*Figure 4B*), we first compute the shape tensor $S_\alpha = \frac{1}{Z_\alpha}\sum_{i=1}^{Z_\alpha}\vec{z}_\alpha^i \otimes \vec{z}_\alpha^i$ where $\vec{z}_\alpha^i$ is the vector from the cell centroid to vertex i (*Nestor-Bergmann et al., 2018*). The shape tensor comes with two positive eigenvalues $\lambda_a$ and $\lambda_b$ and their corresponding eigenvectors.

The cell elongation parameter is computed by $max(\lambda_a, \lambda_b)/min(\lambda_a, \lambda_b)$. The angle deviation is quantified between the eigenvector of $max(\lambda_a, \lambda_b)$ and the local radial vector.

## Stress and strain analysis

We compute the average stress and strain on each Voronoi-cell by the following procedure. The average stress on the cell $\alpha$ is calculated as

$$\sigma^{\alpha} = \frac{1}{A^{\alpha}} \int_{\partial\Omega^{\alpha}\cup\Omega^{\alpha}} \int \sigma dA = \frac{1}{A^{\alpha}} \sum_{\beta=1}^{Z^{\alpha}} \frac{\Lambda_{\beta}^{\alpha}}{2} \hat{t}_{\beta}^{\alpha} \otimes \hat{t}_{\beta}^{\alpha}]_{\beta}^{\alpha} + K\left(\frac{A^{\alpha}}{g^{\alpha}A_{0}^{\alpha}} - 1\right)I$$

where $\hat{t}_{\beta}^{\alpha}$ denotes the unit vector along the junction $\beta$. $Z_{\alpha}$ denotes the total number of junctions (or vertices) of the cell (**Batchelor, 1970**). The radial and circumferential stresses (**Figure 4—figure supplement 3**) are computed by $\hat{r} \cdot \sigma^{\alpha}\hat{r}$ and $\hat{\theta} \cdot \sigma^{\alpha}\hat{\theta}$, respectively, where $\hat{r}$ is the local unit radial vector and $\hat{\theta}$ is the local unit tangential vector.

For the strain, we rescale the two eigenvalues of the shape tensor $S_{\alpha}$ by $\sqrt{A^{\alpha}/\pi\lambda_{a}\lambda_{b}}$ such that the rescaled $\lambda_{a}$ and $\lambda_{b}$ represent the two principal axes of the ellipses with cell area $A^{\alpha}$. By assuming the stress-free cell configuration as a circle with area $gA_{0}^{\alpha}$, we compute the two principal cell strains of the cell as $\lambda_{a}/\sqrt{gA_{0}^{\alpha}/\pi}$ and $\lambda_{b}/\sqrt{gA_{0}^{\alpha}/\pi}$ (**Figure 4—figure supplement 3**).

The scripts for the Voronoi cell model are available through Github, copy archived at swh:1:rev: c943c5d773d4748b9fbc96c9c846279e04173d12 (**Olaranont and St Pierre, 2021**).

## Statistical analysis

Student's t-test was used when there were two groups. One-way ANOVA and post-hoc Tukey's test were used for three or more groups. Mann–Whitney test was used for data that was found not normally distributed. Data are represented as mean ± s.e.m. The $n$ values are determined using G*Power, using the standard deviation and effect size from pilot experiments, with a type I error rate $\alpha$ = 0.05 and power of 80%. In all cases, the actual $n$ values are significantly larger than the desired $n$ value. All the experiments were repeated at least three times independently (biological replicates), and in each experiment, at least two technical replicates (replicates within each experiment) were used. Samples were randomly allocated into experimental groups in the drug treatment experiments, while masking was not used during group allocation, data collection, and/or data analysis.

## Customized MATLAB scripts

The MATLAB scripts used in the modeling and quantification of this work were made available to the research community.

The Voronoi cell model: https://github.com/nolarano/2D_Voronoi_Cell_Radial_Alignment.

The scripts for determining the fiber angle deviation structure tensor: https://github.com/nolarano/AngleDev_StructureTensor.git.

# Acknowledgements

This work is supported in part by the National Science Foundation (CMMI 1662835 and CMMI 1846866 to Y.S., and DMS 2012330 to M.W), the Department of Mechanical and Industrial Engineering at the University of Massachusetts Amherst, and Department of Mathematical Sciences at Worcester Polytechnic Institute. The Conte Nanotechnology Cleanroom Lab is acknowledged for support in microfabrication. The authors acknowledge the University of Massachusetts Amherst Light Microscopy Core for confocal microscopy and Genomics Resource Laboratory for genomics services. M.W thanks Sam Walcott for insightful discussions.

# Additional information

### Funding

| Funder | Grant reference number | Author |
|---|---|---|
| National Science Foundation | CMMI 1662835 | Yubing Sun |
| National Science Foundation | CMMI 1846866 | Yubing Sun |
| National Science Foundation | DMS 2012330 | Min Wu |
| University of Massachusetts Amherst | | Yubing Sun |

| Worcester Polytechnic Institute | Min Wu |
|---|---|

The funders had no role in study design, data collection and interpretation, or the decision to submit the work for publication.

## Author contributions
Tianfa Xie, Conceptualization, Formal analysis, Investigation, Writing - original draft; Sarah R St Pierre, Software, Formal analysis, Investigation, Methodology, Writing - original draft; Nonthakorn Olaranont, Software, Formal analysis, Investigation, Methodology, Writing - review and editing; Lauren E Brown, Formal analysis, Writing - review and editing; Min Wu, Conceptualization, Software, Formal analysis, Supervision, Funding acquisition, Investigation, Writing - original draft, Project administration; Yubing Sun, Conceptualization, Formal analysis, Supervision, Funding acquisition, Writing - original draft, Project administration

## Author ORCIDs
Tianfa Xie (iD) https://orcid.org/0000-0003-1332-4373
Sarah R St Pierre (iD) https://orcid.org/0000-0001-9774-8709
Min Wu (iD) https://orcid.org/0000-0003-4728-6475
Yubing Sun (iD) https://orcid.org/0000-0002-6831-3383

## Decision letter and Author response
Decision letter https://doi.org/10.7554/eLife.60381.sa1
Author response https://doi.org/10.7554/eLife.60381.sa2

## Additional files

### Supplementary files
• Transparent reporting form

### Data availability
All data generated or analysed during this study are included in the manuscript and supporting files. All sequencing data were uploaded to the GEO public repository (https://www.ncbi.nlm.nih.gov/geo/) and were assigned series (GSE148155).

The following dataset was generated:

| Author(s) | Year | Dataset title | Dataset URL | Database and Identifier |
|---|---|---|---|---|
| Xie T, Pierre SR, Sun Y | 2020 | Cell contractility and an actin gradient drive polar alignment of fibroblasts in constrained geometries | https://www.ncbi.nlm.nih.gov/geo/query/acc.cgi?acc=GSE148155 | NCBI Gene Expression Omnibus, GSE148155 |

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
