## [Decision Letter]

**Acceptance summary:**

The manuscript combines an impressive array of experimental and modeling approaches to study cell morphological changes due to stiffness heterogeneities and contractility. The article is an interesting, well-written contribution to the field, with the discussion and conclusion well supported by the experimental data.

**Decision letter after peer review:**

Thank you for submitting your article "Condensation tendency of connected contractile tissue with planar isotropic actin network" for consideration by *eLife*. Your article has been reviewed by 3 peer reviewers, and the evaluation has been overseen by a Reviewing Editor and Aleksandra Walczak as the Senior Editor.

The reviewers have discussed the reviews with one another and the Reviewing Editor has drafted this decision to help you prepare a revised submission.

This article reports the radial alignment of rat embryonic fibroblasts at the periphery of circular confinement patterns. It combines a large array of experimental and modeling approaches to study the origin of this phenomenon and find that contractility, adhesion and stiffness gradient are necessary to obtain this alignment.

Summary:

The authors study the effect of confinement on the alignment of REF cells confined within circular micropatterned islands. They observed that the cells are aligned perpendicularly to the boundary after 48h, contrary to other elongated cells such as NIH-3T3. After testing several subclones of that cell line, they identified cell contractility and cell-cell adhesion affect the organization of the cells in the circular patterns. They confirmed this finding using drugs that affect contractility and disrupt cell adhesion. Then they compared their results to a continuum model and to a Voronoi model.

Enthusiasm for the work is diminished by the limited experimental support for key assumptions of the conceptual and math models (e.g. existence of stiffness gradient, assumption of uniform contractility, use of calcium chelator to show importance of adhesion). Further, integration of model and experiment could be improved, and some of the narrower assumptions of the models (e.g. omitting cell proliferation, remodeling of cell-cell contacts, and cell-substrate interactions, assuming uniform contractility) need better justification. Also, a clear correlate to specific events in development, physiology, or disease would highlight the broader impact of the work beyond a very specific event in a carefully engineered system. Finally, 3 similar papers came out on arxiv from the Roux group. They should be discussed in the manuscript and cited.

Essential revisions:

1. Several assumptions underlying the models need substantiation.

a) The assumption of a purely elastic process: Figure 1A show a dramatic increase in the number of REF2c cells from 24 to 48 hours, suggesting that cells are proliferating. This, together with continuous remodeling of cell-cell contacts, would result in deformations that dissipate elastic energy. Neither modeling approach accounts for this. It would be important for authors incorporate these behaviors, or to provide evidence that cell proliferation and remodeling are unimportant, and similar between the three cell populations being compared.

b) The assumption that contractility is uniform: Work cited (Tambe et al) shows on the contrary that collective cell behaviors exhibit highly heterogeneous active stresses. Experimentally, there are a few potential ways to clarify this point. The authors could use the stiffer (1 MPa) micro post cultures, which recreate radial alignment seen on micropatterned PDMS islands, and compute force variations from post deflection. Alternatively, the authors could perform short time lapse experiments to measure deformations following treatment with blebbistatin or Y27632. Yet another option would be to perform staining for contractile proteins such as phospho-myosin light chain, GTP-bound RhoA, or others, to confirm they are uniformly distributed despite the heterogeneity of F-actin (although such experiments might not reveal uniform contractility when F-actin is nonuniform). Finally, if no experimental support is possible, then authors could turn to model simulations to test whether spatial heterogeneities in contractility alter the overall behavior of the system. In addition to either modeling or experimental support for the assumption that contractility is uniform, authors should provide examples from the literature on related systems that support this assumption.

c) The importance of a stiffness gradient in the cell population, which is one of the key aspects of this work: evidence for the existence of such a gradient is provided only by staining for F-actin, which is insufficient. While F-actin is indeed a key cytoskeletal component in defining the stiffness of cells, the link between intensity of staining and stiffness needs to be proven. Only a single reference is provided, which focused on one specific cancer cell line and the role of stress fibers in stiffening the cell. Moreover, given that F-actin interacts with nonmuscle myosin to form the key contractile machinery of most cell types, heterogeneity in F-actin likely implies heterogeneity in contractility as well. There are also concerns with the measurement of F-actin abundance, including need for statistics on the spatial distribution, and to normalize per cell to reflect variations in F-actin as opposed to simply variations in cell density, which are also present (Figure 1A). Finally, the F-actin gradient is only shown and quantified when intensities are summed over many samples. It would be important to demonstrate a significant gradient within individual samples, and how it varies across samples.

2. The importance of cell-cell adhesion is another crux of the story, pointing to differences underlying the various polarization phenotypes. However, the only experimental support for this is via treatment with a calcium chelator, EGTA. Only one reference is provided for this method (#35, Chen et al), yet Chen et al. appear not to have used EGTA at all, and instead disrupted E-Cadherin using neutralizing antibodies. This is a much more specific and direct approach that the authors of the present study should consider in place of EGTA. In the absence of this or similarly targeted approaches (RNAi, etc), author should include control experiments that demonstrate this rather broad perturbation does not alter contractility or cell-substrate interactions. This could be done at least in part, by using the traction force measurement system the authors have devised. It is particularly important to do so given the importance of calcium for cytoskeletal contraction via calmodulin. A second experiment authors could supplement this with is pharmacologic inhibition of calcium-depdendent contractility, with the hope/expectation that calmodulin-mediated contractility does not predominate this system. Even with these experiments, however, the authors need to provide support from published work that this method of disrupting cell-cell adhesion is well established.

3. The relationship between the in vitro system used by the authors and in vivo phenomena is weak. This is particularly true for the continuum model, where it is nontrivial to relate strain and stress to cell shape changes, given that cell shape is not simply an affine elastic deformation owing to stresses acting on it, but instead a response to stresses integrated with cell autonomous behaviors. There is a large body of literature on the alignment of cells relative to the direction of applied static or dynamic stretch. This mechano-responsivity that dictates cell shape is not considered in the present study. Even without considering these complicating cell behaviors, it is not clear how the magnitude of stress or strain relate to the change in cell shape. In addition, authors would ideally make use of the models to pinpoint what underlies the distinct polarization phenotypes between REF2c, REF11, and 3T3 cell types. The in vitro system should be used to measure directly the cell generated forces.

4. Some terms are either not properly defined or misused and the writing is sometimes unclear. What is the "condensation" process the authors are referring to and how this is related to the boundary alignment of the REF cells. It is the first word in the title of the manuscript, still it appears for the first time in the text only on page 18, where it is poorly defined. Please, read the work of Trepat et al. on active dewetting published in 2018. What do the authors mean by "tendency" (sometimes there is condensation, sometime there isn't?). A different wording to explain their results might be better. Furthermore, terms like nematic, or symmetry are misused.

5. There is a lot of data, analysis and model, but their presentation is not well organized, such that serious rewriting and re-organizing seems in order. The authors chose to show all the analysis they could do in the figures, and therefore there is no clear take-home message. Are all those plots necessary? What is the main result? We suggest to focus on the essential findings. Also, plot the 2 cells types in the same graph instead of showing one graph/cell type.

[Editors' note: further revisions were suggested prior to acceptance, as described below.]

Thank you for submitting your article "Condensation tendency and planar isotropic actin gradient induce radial alignment in confined monolayers" for consideration by *eLife*. Your article has been reviewed by 3 peer reviewers, and the evaluation has been overseen by Aleksandra Walczak as the Senior and Reviewing Editor. The reviewers have opted to remain anonymous.

Essential revisions:

Please answer the Reviewers comments. They are mainly discussion points (you do not need to do new experiments), but please do so carefully.

*Reviewer #1:*

The authors were extremely responsive to reviewer critiques, including text changes, new experiments, and modeling simulation. There are some key places where concerns remain, and these should be addressed via text changes, explicitly describing limitations in the Discussion. Specifically:

1) The modeling of cell responses as purely elastic in the continuum model remains problematic. While new data indicates that cell alignment does not require proliferation, authors do not address the extent to which energy is dissipated via remodeling of cell-cell junctions. This is accounted for in the Voronoi model, as author's point out, but it's absence from the continuum model is a strong limitation that should be noted.

2) Most critically, there remains only indirect support for two key assumptions of the work. One that there is a stiffness gradient, inferred from actin staining, and the other that contractility is uniform and/or unimportant. Given that contraction is produced by interaction of myosin and actin, homogeneous myosin and graded actin can still produce heterogeneous contractile forces. The authors have performed some simulations to test a role for contractility but this is within a modeling framework built on the interpretation of authors, so to some extent you get out what you put in. It would have been ideal to see inhibitor experiments that block myosin activity along with model predictions of the resulting phenotype. I'm the absence of this, some concession on these assumptions needs to be articulated in the Discussion.

*Reviewer #2:*

This submission is a revised manuscript on the radial alignment of REF cells at the periphery of circular confinement patterns. The revisions are significant, and address the comments raised by the reviewers with a number of new experiments and analyses. I recommend publication in *eLife*.

*Reviewer #3:*

I'd like to thank the authors for their corrections. I think the manuscript has improved in clarity significantly. Still, some of the main comments remain unaddressed.

I am summarizing below the main concerns from the last round of reviews and whether they were addressed or not.

1. Justification of the model assumptions:

– Existence of stiffness gradient: ok (thanks to more citations);

– Calcium chelation: ok (thanks to more citations);

– No cell proliferation: ok (thanks to new experiments);

– Uniform contractility: NOT ok (not enough evidence, details below).

2. Integration of model and experiment: No changes.

3. Connection to development, physiology, or disease: No changes.

4. Discussion of the Roux papers: *not* ok.

I am now giving more details on some of the main points that were not adequately addressed:

1. Uniform contractility: The authors performed new immunostaining and new simulations. They showed that the density of myosin motors was uniform on most of the tissue, with a slight increase on the boundary cells. they then performed simulations to show that this slight increase of contractility at the edge does not lead to radial alignment. However, this is not enough experimental evidence to conclude that contractility is uniform. indeed, uniform myosin + non-uniform Factin does not necessarily mean uniform contractility as both proteins are required for contraction.

The authors show in simulations that an increase of contractility at the edge leads to smaller boundary cells. What would happen if contractility follows the actin gradient? Could that be enough to reproduce the radial alignment of the cells, even in the absence of cell contractility differential or condensation?

2. Integration of model and experiments. there is no quantitative comparison between theory and experiments. The simulation results are summarized in the new panel Figure 4B. Instead, why not showing the same radial profiles as for the experiments? It will make the comparison between experiments and theory easier.

It would be neat to see how sensitive the cell alignment is to rho, g and their gradients.

Along those lines, can the authors comment on how steep the stiffness gradient has to be in order to re-align the cells radially? Can this be quantitatively compared to the actin gradients measured in Figure S3-1? Not all the patches seem to display the gradient (see images in the 3rd column). Can the authors show if the steepness of the actin gradient correlates with the radial alignment of the cells or their size for each pattern instead of showing that the average gradient value and the average alignment are correlated?

3. Discussion of the 3 papers from Aurelien Roux's lab: The papers are cited in the bulk of the introduction, but not properly discussed. It seems to me that the transition from parallel to perpendicular anchoring is particularly important to mention in this manuscript. The authors claim in their response that they discussed the papers in the discussion. I could not find this section (none of the 3 papers are cited in the Discussion section). I think comparing and contrasting the results of those papers with this manuscript would greatly improve our understanding of how contractility and circular confinement impact the organization of dense layers of elongated cells.

---

## [Author Response]

Summary:The authors study the effect of confinement on the alignment of REF cells confined within circular micropatterned islands. They observed that the cells are aligned perpendicularly to the boundary after 48h, contrary to other elongated cells such as NIH-3T3. After testing several subclones of that cell line, they identified cell contractility and cell-cell adhesion affect the organization of the cells in the circular patterns. They confirmed this finding using drugs that affect contractility and disrupt cell adhesion. Then they compared their results to a continuum model and to a Voronoi model.Enthusiasm for the work is diminished by the limited experimental support for key assumptions of the conceptual and math models (e.g. existence of stiffness gradient, assumption of uniform contractility, use of calcium chelator to show importance of adhesion). Further, integration of model and experiment could be improved, and some of the narrower assumptions of the models (e.g. omitting cell proliferation, remodeling of cell-cell contacts, and cell-substrate interactions, assuming uniform contractility) need better justification. Also, a clear correlate to specific events in development, physiology, or disease would highlight the broader impact of the work beyond a very specific event in a carefully engineered system. Finally, 3 similar papers came out on arxiv from the Roux group. They should be discussed in the manuscript and cited.

As the reviewers will see below, we have done more experiments and quantifications to support the existence of the stiffness gradient, the assumption of uniform contractility, and the importance of coherent cell-cell adhesions. Our model focuses on the mechanical equilibrium when the dynamics of cell-cell contact exchange and cell proliferation have ceased. Additionally, we have further shown the radial alignment is not caused directly by proliferation and mechanotransduction. We believe the new experiments and simulations have pinned our major conclusion that in the connected contractile REF tissue, the radial-alignment boundary is due to the combination of the condensation tendency, which describes the collective cell behavior that cells condense on soft substrates, and cell stiffness differential.

In multiple developmental and physiological processes, previous focuses have been on the differential growth/contractility as a tissue flow-and-deformation driver^1, 2^ while there is relatively little evidence (except in *Drosophila* development^3, 4^) that the differential stiffness also plays an important role. In 2D tissues, it is difficult to disentangle differential contractility and stiffness as the two mechanical properties are both connected to the actomyosin activities. Using engineered systems, we have disentangled both experimentally and theoretically their roles in regulating the cell morphology at the REF tissue boundary and we reason it is the differential stiffness rather than the differential contractility that contributes to the radial-alignment boundary. While engineered patterns help the quantitative study of the cell behaviors, we believe the condensation tendency and the emerging stiffness gradient are intrinsic tissue properties that contribute to tissue morphogenesis. For example, when epithelial cells undergo epithelial-mesenchymal transition, the actin stress fiber distribution starts to change while maintaining functional cell-cell junctions. It is possible that local condensations can be observed in this process. Also, neural crest cells and chondrocytes migrate collectively and form aggregates during craniofacial and tooth development, respectively. It is also possible that these mesenchymal condensation processes are also regulated by cell-stiffness differential and condensation tendency, in addition to chemoattractant as previously thought. We have highlighted these potential broader impacts in the discussion of the revised manuscript.

We appreciate the reviewers for pointing out the important recent works by Roux group. We have cited and discussed these 3 papers in the introduction and discussion^5-7^.

Essential revisions:1. Several assumptions underlying the models need substantiation.a) The assumption of a purely elastic process: Figure 1A show a dramatic increase in the number of REF2c cells from 24 to 48 hours, suggesting that cells are proliferating. This, together with continuous remodeling of cell-cell contacts, would result in deformations that dissipate elastic energy. Neither modeling approach accounts for this. It would be important for authors incorporate these behaviors, or to provide evidence that cell proliferation and remodeling are unimportant, and similar between the three cell populations being compared.

We agree that the cell proliferation from 24 to 48 hrs may lead to deformations that dissipate elastic energy. To evaluate whether cell proliferation changes the cell alignment results, we treated REF2c cells with aphidicolin, an inhibitor of DNA replication. We found that with aphidicolin treatment, the cell proliferation was significantly inhibited, as reflected by a non-significant cell number increase. In this condition, we still observed the radial alignment of boundary cells, which is quantitatively comparable with untreated controls. Thus, we believe that the cell proliferation mediated remodeling is unimportant for the formation of cell radial alignment (Figure 4—figure supplement 1).

We have developed a Voronoi model to explain the cell radial alignment in the mechanical equilibrium, with stiffness differential between the inner and boundary cells and cell contractility. This equilibrium is due to the minimization of the tissue elastic energy. During the relaxation of the tissue energy, cell centroids move along the direction that decreases (dissipates) the total energy, where cell-cell contact may change.

b) The assumption that contractility is uniform: Work cited (Tambe et al) shows on the contrary that collective cell behaviors exhibit highly heterogeneous active stresses. Experimentally, there are a few potential ways to clarify this point. The authors could use the stiffer (1 MPa) micro post cultures, which recreate radial alignment seen on micropatterned PDMS islands, and compute force variations from post deflection. Alternatively, the authors could perform short time lapse experiments to measure deformations following treatment with blebbistatin or Y27632. Yet another option would be to perform staining for contractile proteins such as phospho-myosin light chain, GTP-bound RhoA, or others, to confirm they are uniformly distributed despite the heterogeneity of F-actin (although such experiments might not reveal uniform contractility when F-actin is nonuniform). Finally, if no experimental support is possible, then authors could turn to model simulations to test whether spatial heterogeneities in contractility alter the overall behavior of the system. In addition to either modeling or experimental support for the assumption that contractility is uniform, authors should provide examples from the literature on related systems that support this assumption.

We thank the reviewers for this insightful comment. We combined experiments and simulations to investigate whether the contractility was uniform within the pattern and the importance of the uniformity of the contractility to our model. To evaluate the total contractility, it is not accurate to only measure traction forces due to strong cell-cell adhesions. In addition, as shown in Figure 1—figure supplement 5, the traction force measurement is challenging using soft PDMS micropost arrays (8.4 µm) due to the condensation of the cell colony. It is not possible to use stiffer micropost arrays to measure traction force, as the deflections are too small to be measured accurately^8^. Thus, as suggested by the reviewers, we performed staining for phosphorylated-myosin (p-myosin) light chain to investigate the distribution of p-myosin (see Figure 3—figure supplement 3). We found that although F-actin gradients existed within the patterns, the p-myosin distribution was more uniform, with only a slight increase near the pattern boundary, which was more significant when normalized by cell area, as the boundary cells were larger than inner cells. This is opposite to the increasing F-actin intensity towards the pattern center.

We further performed simulations in the Voronoi cell model and have found that the contractility gradient suggested by the image data of phosphorylated-myosin distribution could not result in the radial alignment of boundary cells (Figure 4, figure supplement 2). Rather, only smaller cell areas at the border were found, which contradicts the experimental observation in REF2c tissues. Combining both old and new simulation results, we further confirm that the actin gradient is responsible for the radial alignment of boundary cells.

c) The importance of a stiffness gradient in the cell population, which is one of the key aspects of this work: evidence for the existence of such a gradient is provided only by staining for F-actin, which is insufficient. While F-actin is indeed a key cytoskeletal component in defining the stiffness of cells, the link between intensity of staining and stiffness needs to be proven. Only a single reference is provided, which focused on one specific cancer cell line and the role of stress fibers in stiffening the cell. Moreover, given that F-actin interacts with nonmuscle myosin to form the key contractile machinery of most cell types, heterogeneity in F-actin likely implies heterogeneity in contractility as well. There are also concerns with the measurement of F-actin abundance, including need for statistics on the spatial distribution, and to normalize per cell to reflect variations in F-actin as opposed to simply variations in cell density, which are also present (Figure 1A).Finally, the F-actin gradient is only shown and quantified when intensities are summed over many samples. It would be important to demonstrate a significant gradient within individual samples, and how it varies across samples.

We provided more references to support the connection between F-actin intensity and cell stiffness^9-11^. The heterogeneity in contractility has been addressed in our response to comment 1b above. As suggested, we further performed statistical analysis by dividing the pattern into 6 regions and compared average actin intensity within each zone. As shown in Figure 3, figure supplement 2A, a significant decrease was found in the outmost layer. To exclude the possibility that this drop in actin intensity is due to decreased cell density in the pattern boundary, we compared both average actin intensity (Figure 3, figure supplement 2B) and average actin intensity per cell (Figure 3, figure supplement 2B) between inner cells and boundary cells (cells in the outmost layer). The statistical analysis results supported that a significant decrease in actin intensity could still be found after normalizing to cell density.

As suggested by the reviewers, we now include three representative fluorescence images and actin intensity profiles showing the actin gradient within individual patterns (Figure 3, figure supplement 1). These images further confirmed such actin intensity gradient could be found within each individual pattern.

2. The importance of cell-cell adhesion is another crux of the story, pointing to differences underlying the various polarization phenotypes. However, the only experimental support for this is via treatment with a calcium chelator, EGTA. Only one reference is provided for this method (#35, Chen et al), yet Chen et al. appear not to have used EGTA at all, and instead disrupted E-Cadherin using neutralizing antibodies. This is a much more specific and direct approach that the authors of the present study should consider in place of EGTA. In the absence of this or similarly targeted approaches (RNAi, etc), author should include control experiments that demonstrate this rather broad perturbation does not alter contractility or cell-substrate interactions. This could be done at least in part, by using the traction force measurement system the authors have devised. It is particularly important to do so given the importance of calcium for cytoskeletal contraction via calmodulin. A second experiment authors could supplement this with is pharmacologic inhibition of calcium-dependent contractility, with the hope/expectation that calmodulin-mediated contractility does not predominate this system. Even with these experiments, however, the authors need to provide support from published work that this method of disrupting cell-cell adhesion is well established.

We apologize for this wrong reference and replaced it with the correct one^12^. In Figure 4D-F of this paper, Ohgushi et al. used EGTA to “disrupts the cadherin-mediated cell attachment and caused hESC to detach from one another but not from the plate bottom”. It is challenging to design targeted inhibition experiments, because unlike epithelial cells in most previous studies, it is unclear what types of cadherin the REF2c cells express, and thus we chose to use a calcium chelator to universally inhibit all cadherin-mediated cell-cell adhesion.

To further confirm that EGTA treatment mainly regulates cell-cell interactions but not cell-substrate interactions, we measured the contractility of cells using the PDMS micropost arrays as suggested by the reviewers. We found that the EGTA treatment did not significantly change cell contractility (see Figure 2—figure supplement 3), suggesting calmodulin-mediated contractility does not predominate this system. As suggested by the reviewers, we also provided more references to support the use of EGTA to disrupt cell-cell adhesions. For example, in the paper mentioned several times by reviewers (Pérez-González et al., 2019), EGTA was used to disrupt cell-cell adhesion^13^. Others have also reported using EGTA to disrupt E-cadherin dependent cell-cell adhesions in MDCK cells^14^ and S180 cells^15^, and N-cadherin dependent cell-cell adhesions in vascular smooth muscle cell^16^ and C2C12 myoblasts^17^. We believe those works are sufficient to support that EGTA has been widely used to disrupt cell-cell adhesions.

3. The relationship between the in vitro system used by the authors and in vivo phenomena is weak. This is particularly true for the continuum model, where it is nontrivial to relate strain and stress to cell shape changes, given that cell shape is not simply an affine elastic deformation owing to stresses acting on it, but instead a response to stresses integrated with cell autonomous behaviors. There is a large body of literature on the alignment of cells relative to the direction of applied static or dynamic stretch. This mechano-responsivity that dictates cell shape is not considered in the present study. Even without considering these complicating cell behaviors, it is not clear how the magnitude of stress or strain relate to the change in cell shape. In addition, authors would ideally make use of the models to pinpoint what underlies the distinct polarization phenotypes between REF2c, REF11, and 3T3 cell types. The in vitro system should be used to measure directly the cell generated forces.

We thank the reviewers for bringing up this important issue. It is well-documented that various types of cells respond to mechanical stretching by aligning perpendicular to the stretching directions. As pointed out by the reviewers, complex mechanotransduction pathways, including focal adhesion mediated signaling^18-20^ and YAP-TEAD signaling^21-23^. In our case, we believe the cell shape changes are modeled as the direct responses to material properties changes of cells due to actin remodeling. This does not exclude the possibility that cells actively respond to mechanical forces and remodel the actin cytoskeleton. To further examine the activation of mechanotransduction signals in our system, we stained focal adhesion proteins vinculin (Figure 4—figure supplement 4) and canonical mechanosensor YAP at 48 hr when radial alignment is prominent (Figure 4—figure supplement 5). If mechanotransduction is involved, we expect to see a higher intensity of vinculin and/or nuclear YAP in the boundary cells. However, we found that the distribution of vinculin was mostly uniform across the pattern area, except for reduced signals near the pattern boundary. Notably, the overall vinculin intensity per cell in the outmost layer of cells was still comparable with inner cells. We also found that most of the REF2c cells expressed nuclear YAP, which is independent of confinement and cell alignment status (24 hr vs. 48 hr). These results suggest local activation of these mechanotransduction pathways in boundary cells is unlikely the cause for its radial alignment.

As suggested by the reviewers, we further used our model to reveal the quantitative relationship between the magnitude of stress or strain and the change in cell shape and orientation. We agree with the reviewers that it is non-trivial to connect cell shape with cell strain in both the Voronoi model and the continuum model, since the reference stress-free shape of the cell is not known. Nevertheless, due to the isotropic orientation of actin, we assume that the reference stress-free configuration of the cell is isotropic and the reference cell size (area) is determined by the level of contractility (see Stress and Strain Analysis in the Materials and methods). Then we can estimate the cell stress and strain under different levels of stiffness gradient and contractility, and their connection with cell shape and orientation. (Figure 4 and Figure 4—figure supplement 3, orange dots: boundary cells; blue dots: inner cells). REF2c is represented by contractile cells with stiffness gradient (e.g., *g* = 0.5 and ρ=0.4) and REF11b is represented by non-contractile cells with stiffness gradient (e.g., *g* = 1 and ρ=0.4). We think that the 3T3 cell pattern cannot be simulated by our modeling approaches where the REF tissue is clearly a continuum. Since there is no clear junction between 3T3 cells (see β-catenin in Figure 2—figure supplement 2), the dominant mechanical interaction can be very different between 3T3 and REF cells.

We agree with the reviewers that a direct measurement of traction force distribution is critical. However, as we have shown in Figure 1—figure supplement 5, a strong tendency of condensation of PDMS micropost arrays with an effective modulus of 5 kPa was observed, making it challenging to measure the traction forces, as the post deflection will be too small to be measured accurately if using shorter, and thus stiffer, PDMS microposts^8^.

4. Some terms are either not properly defined or misused and the writing is sometimes unclear. What is the "condensation" process the authors are referring to and how this is related to the boundary alignment of the REF cells. It is the first word in the title of the manuscript, still it appears for the first time in the text only on page 18, where it is poorly defined. Please, read the work of Trepat et al. on active dewetting published in 2018. What do the authora mean by "tendency" (sometimes there is condensation, sometime there isn't?). A different wording to explain their results might be better. Furthermore, terms like nematic, or symmetry are misused.

We thank the reviewers to point this term out and refer our work to Pérez-González et al., 2019. In our system, we observed this radial alignment of boundary cells when REF2c cells were confined on rigid substrates, driven by an actin gradient concentrated in the pattern center. As shown in Figure 1—figure supplement 5, we observed a clear increase of cell density in the pattern center on PDMS micropost arrays substrates with lower stiffness. This is also a reminiscence of the mesenchymal condensation phenomenon, which is a prevalent morphogenetic transition that involves the aggregation of mesenchymal cells^24-26^. We used the term “tendency” to describe the fact that when cells were cultured on flat PDMS substrates, they reached an equilibrium state without significant cell motions, so the cells do not actually “condense” to the center (while they indeed condensed on soft substrates). We would like to describe this emergent collective behavior of REF2c cells and differentiate it from conventional mesenchymal condensation.

It is notable that our observation, to some extent, is indeed similar to the dewetting process in the epithelial tissue in Pérez-González et al., 2019. While Pérez-González et al., 2019 aimed to explain the out-of-equilibrium process by describing the tissue as active viscous fluid, we are trying to explain the cell alignment in mechanical equilibrium when the tissue is confined in a stiffer substrate, which is not observed in the epithelial tissue. Moreover, the active viscous fluid theory developed in Pérez-González et al., 2019, where active forces drive the tissue flow cannot explain a final tissue pattern in static equilibrium, but rather the initial onset of tissue retraction. In the revised manuscript, we have provided a detailed explanation of the condensation tendency at the beginning of the manuscript, and referred to Pérez-González et al., 2019, and discuss the similarity in the strong contractility plus cell-cell junction and difference in equilibrium state and out-of-equilibrium process.

5. There is a lot of data, analysis and model, but their presentation is not well organized, such that serious rewriting and re-organizing seems in order. The authors chose to show all the analysis they could do in the figures, and therefore there is no clear take-home message. Are all those plots necessary? What is the main result? We suggest to focus on the essential findings. Also, plot the 2 cells types in the same graph instead of showing one graph/cell type.

We have reorganized the figures and the manuscript to focus on essential findings in the main figures and moving supporting results to the supplementary information. Specifically, we removed the continuum model results (original Figure 4) and simulation results for the ring patterns (Original Figure 6). We also moved the ring pattern experimental results (original Figure 7) to supplementary figures. In the Voronoi cell model (original Figure 5), we moved original bar plots showing statistical analysis (original Figure 5C-D) to supplementary figures and added scatter plots showing angle deviation, elongation, and cell area, and generate new scatter plots to show the changes of cell shape in relation to stress and strains (see the new Figure 4). We believe these substantial changes help us to focus on the essential findings. In all the figures, we presented results from various cell types in parallel.

References:

1. Ambrosi D, Ben Amar M, Cyron CJ, DeSimone A, Goriely A, Humphrey JD, Kuhl E. Growth and remodelling of living tissues: perspectives, challenges and opportunities. J R Soc Interface. 2019;16(157).

2. Streichan SJ, Lefebvre MF, Noll N, Wieschaus EF, Shraiman BI. Global morphogenetic flow is accurately predicted by the spatial distribution of myosin motors. *eLife*. 2018;7:e27454.

3. Polyakov O, He B, Swan M, Shaevitz JW, Kaschube M, Wieschaus E. Passive mechanical forces control cell-shape change during *Drosophila* ventral furrow formation. Biophys J. 2014;107(4):998-1010.

4. Rauzi M, Krzic U, Saunders TE, Krajnc M, Ziherl P, Hufnagel L, Leptin M. Embryo-scale tissue mechanics during *Drosophila* gastrulation movements. Nat Commun. 2015;6:8677.

5. Blanch-Mercader C, Guillamat P, Roux A, Kruse K. Integer topological defects of cell monolayers: Mechanics and flows. Phys Rev E. 2021;103(1-1):012405.

6. Blanch-Mercader C, Guillamat P, Roux A, Kruse K. Quantifying Material Properties of Cell Monolayers by Analyzing Integer Topological Defects. Phys Rev Lett. 2021;126(2):028101.

7. Guillamat P, Blanch-Mercader C, Kruse K, Roux A. Integer topological defects organize stresses driving tissue morphogenesis. bioRxiv. 2020:2020.06.02.129262.

8. Fu JP, Wang YK, Yang MT, Desai RA, Yu XA, Liu ZJ, Chen CS. Mechanical regulation of cell function with geometrically modulated elastomeric substrates. Nat Meth. 2010;7(9):733-6.

9. Rotsch C, Radmacher M. Drug-induced changes of cytoskeletal structure and mechanics in fibroblasts: an atomic force microscopy study. Biophys J. 2000;78(1):520-35.

10. Nawaz S, Sánchez P, Bodensiek K, Li S, Simons M, Schaap IAT. Cell Visco-Elasticity Measured with AFM and Optical Trapping at Sub-Micrometer Deformations. PLOS ONE. 2012;7(9):e45297.

11. Gavara N, Chadwick RS. Relationship between cell stiffness and stress fiber amount, assessed by simultaneous atomic force microscopy and live-cell fluorescence imaging. Biomech Model Mechanobiol. 2016;15(3):511-23.

12. Ohgushi M, Matsumura M, Eiraku M, Murakami K, Aramaki T, Nishiyama A, Muguruma K, Nakano T, Suga H, Ueno M, Ishizaki T, Suemori H, Narumiya S, Niwa H, Sasai Y. Molecular Pathway and Cell State Responsible for Dissociation-Induced Apoptosis in Human Pluripotent Stem Cells. Cell Stem Cell. 2010;7(2):225-39.

13. Perez-Gonzalez C, Alert R, Blanch-Mercader C, Gomez-Gonzalez M, Kolodziej T, Bazellieres E, Casademunt J, Trepat X. Active wetting of epithelial tissues. Nat Phys. 2019;15(1):79-88.

14. Rothen-Rutishauser B, Riesen FK, Braun A, Gunthert M, Wunderli-Allenspach H. Dynamics of tight and adherens junctions under EGTA treatment. J Membr Biol. 2002;188(2):151-62.

15. Al-Kilani A, de Freitas O, Dufour S, Gallet F. Negative feedback from integrins to cadherins: a micromechanical study. Biophys J. 2011;101(2):336-44.

16. Koutsouki E, Beeching Cressida A, Slater Sadie C, Blaschuk Orest W, Sala-Newby Graciela B, George Sarah J. N-Cadherin–Dependent Cell–Cell Contacts Promote Human Saphenous Vein Smooth Muscle Cell Survival. Arteriosclerosis, Thrombosis, and Vascular Biology. 2005;25(5):982-8.

17. Charrasse S, Meriane M, Comunale F, Blangy A, Gauthier-Rouvière Cc. N-cadherin–dependent cell–cell contact regulates Rho GTPases and β-catenin localization in mouse C2C12 myoblasts. Journal of Cell Biology. 2002;158(5):953-65.

18. Greiner AM, Chen H, Spatz JP, Kemkemer R. Cyclic tensile strain controls cell shape and directs actin stress fiber formation and focal adhesion alignment in spreading cells. PLoS One. 2013;8(10):e77328.

19. Greiner AM, Biela SA, Chen H, Spatz JP, Kemkemer R. Temporal responses of human endothelial and smooth muscle cells exposed to uniaxial cyclic tensile strain. Exp Biol Med (Maywood). 2015;240(10):1298-309.

20. Qian J, Liu H, Lin Y, Chen W, Gao H. A mechanochemical model of cell reorientation on substrates under cyclic stretch. PLoS One. 2013;8(6):e65864.

21. Ugolini GS, Rasponi M, Pavesi A, Santoro R, Kamm R, Fiore GB, Pesce M, Soncini M. On-chip assessment of human primary cardiac fibroblasts proliferative responses to uniaxial cyclic mechanical strain. Biotechnol Bioeng. 2016;113(4):859-69.

22. Aragona M, Sifrim A, Malfait M, Song Y, Van Herck J, Dekoninck S, Gargouri S, Lapouge G, Swedlund B, Dubois C, Baatsen P, Vints K, Han S, Tissir F, Voet T, Simons BD, Blanpain C. Mechanisms of stretch-mediated skin expansion at single-cell resolution. Nature. 2020;584(7820):268-73.

23. Panciera T, Azzolin L, Cordenonsi M, Piccolo S. Mechanobiology of YAP and TAZ in physiology and disease. Nat Rev Mol Cell Biol. 2017;18(12):758-70.

24. Mammoto T, Mammoto A, Torisawa Y-s, Tat T, Gibbs A, Derda R, Mannix R, de Bruijn M, Yung Chong W, Huh D, Ingber Donald E. Mechanochemical Control of Mesenchymal Condensation and Embryonic Tooth Organ Formation. Developmental Cell. 2011;21(4):758-69.

25. Klumpers DD, Mao AS, Smit TH, Mooney DJ. Linear patterning of mesenchymal condensations is modulated by geometric constraints. J R Soc Interface. 2014;11(95):20140215.

26. Lim J, Tu X, Choi K, Akiyama H, Mishina Y, Long F. BMP–Smad4 signaling is required for precartilaginous mesenchymal condensation independent of *Sox9* in the mouse. Developmental Biology. 2015;400(1):132-8.

27. Kranenburg O, Poland M, Gebbink M, Oomen L, Moolenaar WH. Dissociation of LPA-induced cytoskeletal contraction from stress fiber formation by differential localization of RhoA. J Cell Sci. 1997;110 ( Pt 19):2417-27.

28. Toews ML, Ustinova EE, Schultz HD. Lysophosphatidic acid enhances contractility of isolated airway smooth muscle. J Appl Physiol (1985). 1997;83(4):1216-22.

29. Kole TP, Tseng Y, Huang L, Katz JL, Wirtz D. Rho kinase regulates the intracellular micromechanical response of adherent cells to rho activation. Mol Biol Cell. 2004;15(7):3475-84.

30. Duclos G, Erlenkamper C, Joanny JF, Silberzan P. Topological defects in confined populations of spindle-shaped cells. Nat Phys. 2017;13(1):58-62.

31. Takebe T, Enomura M, Yoshizawa E, Kimura M, Koike H, Ueno Y, Matsuzaki T, Yamazaki T, Toyohara T, Osafune K, Nakauchi H, Yoshikawa HY, Taniguchi H. Vascularized and Complex Organ Buds from Diverse Tissues via Mesenchymal Cell-Driven Condensation. Cell Stem Cell. 2015;16(5):556-65.

32. Szabo A, Mayor R. Mechanisms of Neural Crest Migration. Annu Rev Genet. 2018;52:43-63.

[Editors' note: further revisions were suggested prior to acceptance, as described below.]

Essential revisions:Please answer the Reviewers comments. They are mainly discussion points (you do not need to do new experiments), but please do so carefully.Reviewer #1:The authors were extremely responsive to reviewer critiques, including text changes, new experiments, and modeling simulation. There are some key places where concerns remain, and these should be addressed via text changes, explicitly describing limitations in the Discussion. Specifically:1) The modeling of cell responses as purely elastic in the continuum model remains problematic. While new data indicates that cell alignment does not require proliferation, authors do not address the extent to which energy is dissipated via remodeling of cell cell junctions. This is accounted for in the Voronoi model, as author's point out, but it's absence from the continuum model is a strong limitation that should be noted.

As suggested by the reviewers, we have completely deleted the theoretical results related to the continuum pure elastic model in the last revision. In the Discussion section, we have now mentioned that we are currently developing a continuum model to explain the tissue flow before the mechanical equilibrium. This continuum model is a Maxwell-type viscoelastic model, in which the energy dissipation due to the cell-cell junction and cytoskeletal remodeling will be accounted for at the tissue scale. The added discussion reads as follows:

“In particular, a Maxwell-type viscoelastic continuum model will be needed to couple the actin dynamics with active tissue mechanics to explain the onset of tissue flow and the maintenance of the tissue mechanical equilibrium (1). This new framework will account for the energy dissipation due to cell-cell junction and cytoskeletal rearrangement during the tissue flow before equilibrium.”

2) Most critically, there remains only indirect support for two key assumptions of the work. One that there is a stiffness gradient, inferred from actin staining, and the other that contractility is uniform and/or unimportant. Given that contraction is produced by interaction of myosin and actin, homogeneous myosin and graded actin can still produce heterogeneous contractile forces. The authors have performed some simulations to test a role for contractility but this is within a modeling framework built on the interpretation of authors, so to some extent you get out what you put in. It would have been ideal to see inhibitor experiments that block myosin activity along with model predictions of the resulting phenotype. I'm the absence of this, some concession on these assumptions needs to be articulated in the Discussion.

While a positive correlation between the myosin level and contractility is well-documented, as the reviewer suggested, it is not impossible that the increasing actin density can be associated with a higher level of contractility because the actomyosin contractility depends on both the myosin and the architecture of the mesh (2). While we have not found results supporting actin mesh density alone can increase the contractility in the literature, it is still possible because the macroscopic network contraction also depends on the concentration of actin cross-linking proteins. We have modified our discussion on the role of stiffness versus contractility in forming the radial elongation and elaborated more details concerning both possibilities in the discussion, which reads as follows:

“A positive correlation between the actin and myosin spatial distributions has been widely reported previously (3, 4). […] However, this possibility requires further analysis on how myosin motors and actin network molecules interact to generate macroscopic network contraction.”

We indeed have already demonstrated that inhibiting myosin activities (through Blebbistatin and ROCK inhibitors, shown in Figure 2C) led to disruption of radial alignment and attributed this finding to the change of condensation tendency. However, it is challenging to pharmaceutical control myosin gradient within the patterns and thus cannot be used to study the role of myosin gradient. Future works are needed to locally regulate myosin activities.

Reviewer #3:I'd like to thank the authors for their corrections. I think the manuscript has improved in clarity significantly. Still, some of the main comments remain unaddressed.I am summarizing below the main concerns from the last round of reviews and whether they were addressed or not.1. Justification of the model assumptions:– Existence of stiffness gradient: ok (thanks to more citations);– Calcium chelation: ok (thanks to more citations);– No cell proliferation: ok (thanks to new experiments);– Uniform contractility: not ok (not enough evidence, details below).2. Integration of model and experiment: No changes.3. Connection to development, physiology, or disease: No changes.4. Discussion of the Roux papers: not ok.I am now giving more details on some of the main points that were not adequately addressed:1. Uniform contractility: The authors performed new immunostaining and new simulations. They showed that the density of myosin motors was uniform on most of the tissue, with a slight increase on the boundary cells. they then performed simulations to show that this slight increase of contractility at the edge does not lead to radial alignment. However, this is not enough experimental evidence to conclude that contractility is uniform. indeed, uniform myosin + non-uniform Factin does not necessarily mean uniform contractility as both proteins are required for contraction.The authors show in simulations that an increase of contractility at the edge leads to smaller boundary cells. What would happen if contractility follows the actin gradient? Could that be enough to reproduce the radial alignment of the cells, even in the absence of cell contractility differential or condensation?

We thank this reviewer for this insightful comment, which is in line with comment #2 of Reviewer #1. Please see also see our response to that comment.

As pointed out by both reviewers, it is not impossible that the actin gradient also affects the contractility because the actomyosin-based contractility depends on both the myosin at work and the architecture of the actin mesh. In our model, we did not emphasize that uniform contractility is necessary, and we did not exclude the possibility of a contractility gradient associated with the actin. However, since multiple studies are supporting that the stiffness gradient is aligned with the actin gradient, we reason the stiffness differential due to the actin gradient together with the contractility (either graded or not) and intercellular adhesion are sufficient for the radial alignment. We have added more details in the discussion to concern the possibility of a contractility gradient associated with the actin gradient.

2. Integration of model and experiments. there is no quantitative comparison between theory and experiments. The simulation results are summarized in the new panel Figure 4B. Instead, why not showing the same radial profiles as for the experiments? It will make the comparison between experiments and theory easier.

Our model is essential to demonstrate the independent role of condensation tendency and stiffness differential in forming the boundary radial alignment. While we agree that a direct comparison between modeling and experimental results will be interesting, it is difficult to have a fair comparison, mainly because it is difficult to calibrate the prestretch (*g*) in the model in experiments. Therefore, we believe the qualitative comparison between the model and experimental results is suitable in our case.

It would be neat to see how sensitive the cell alignment is to rho, g and their gradients.

These sensitivity studies have already been shown in Figure. 4.

Along those lines, can the authors comment on how steep the stiffness gradient has to be in order to re-align the cells radially? Can this be quantitatively compared to the actin gradients measured in Figure S3-1? Not all the patches seem to display the gradient (see images in the 3rd column). Can the authors show if the steepness of the actin gradient correlates with the radial alignment of the cells or their size for each pattern instead of showing that the average gradient value and the average alignment are correlated?

We thank the reviewer for this suggestion. As the radial alignment of boundary cells is mainly characterized by cell angle deviation, elongation, and cell area, there is not a clear threshold for such alignment. Instead, as suggested by this reviewer, the best way is to investigate the correlation between actin gradient and radial alignment. In that regard, we analyzed a total of 16 colonies. The colonies shown in Figure 3—figure supplement 1A are two typical colonies with different steepness of actin gradient. We found that the bottom one with a steeper actin gradient did show radial alignment more clearly (Figure 3—figure supplement 2A). To quantify this correlation, we generated a scatterplot and performed linear regression analysis and correlation analysis (Figure 3—figure supplement 2B). The results show that angle deviations of boundary cells are negatively correlated with actin gradient with a Pearson correlation coefficient of − 0.533 and a *P* value of 0.0335, indicating a high probability of correlation (smaller angle deviation means better radial alignment). Thus, these results further support that actin gradient is essential for the establishment of boundary cell radial alignment.

3. Discussion of the 3 papers from Aurelien Roux's lab: The papers are cited in the bulk of the introduction, but not properly discussed. It seems to me that the transition from parallel to perpendicular anchoring is particularly important to mention in this manuscript. The authors claim in their response that they discussed the papers in the discussion. I could not find this section (none of the 3 papers are cited in the Discussion section). I think comparing and contrasting the results of those papers with this manuscript would greatly improve our understanding of how contractility and circular confinement impact the organization of dense layers of elongated cells.

We agree with this reviewer that a detailed discussion of the three recent papers from Roux’s group is important. The significant difference between our work and their works is that we have found an equilibrium with the radial elongation pattern while their multicellular system evolves from spiral to radial cell alignment and from radial alignment to out-of-plane tissue growth as cell proliferation continues. We have added a thorough discussion in the revision to compare and contrast the results from our manuscript to the three papers from Aurelien Roux’s lab, which reads as follows:

“Recently, Roux, Kruse, and colleagues reported that C2C12 myoblasts could self-organize into spiral patterns when confined using similar circular confinement (10-12). […] While cell proliferation is a critical driving factor in aster pattern formation in the C2C12 system, in our REF system, the stiffness differential is indispensable for the radial alignment formation.”

References:

1. Chaozhen Wei MW. Nonlinear modeling reveals multi-timescale and higher-order effects in active tissue mechanics. arXiv:210309427. 2021.

2. Kasza KE, Zallen JA. Dynamics and regulation of contractile actin-myosin networks in morphogenesis. Curr Opin Cell Biol. 2011;23(1):30-8.

3. Martin AC, Kaschube M, Wieschaus EF. Pulsed contractions of an actin-myosin network drive apical constriction. Nature. 2009;457(7228):495-9.

4. Banerjee DS, Munjal A, Lecuit T, Rao M. Actomyosin pulsation and flows in an active elastomer with turnover and network remodeling. Nat Commun. 2017;8(1):1121.

5. Murrell M, Oakes PW, Lenz M, Gardel ML. Forcing cells into shape: the mechanics of actomyosin contractility. Nat Rev Mol Cell Biol. 2015;16(8):486-98.

6. Tavares S, Vieira AF, Taubenberger AV, Araujo M, Martins NP, Bras-Pereira C, et al. Actin stress fiber organization promotes cell stiffening and proliferation of pre-invasive breast cancer cells. Nat Commun. 2017;8:15237.

7. Rotsch C, Radmacher M. Drug-induced changes of cytoskeletal structure and mechanics in fibroblasts: an atomic force microscopy study. Biophys J. 2000;78(1):520-35.

8. Nawaz S, Sánchez P, Bodensiek K, Li S, Simons M, Schaap IAT. Cell Visco-Elasticity Measured with AFM and Optical Trapping at Sub-Micrometer Deformations. PLOS ONE. 2012;7(9):e45297.

9. Gavara N, Chadwick RS. Relationship between cell stiffness and stress fiber amount, assessed by simultaneous atomic force microscopy and live-cell fluorescence imaging. Biomech Model Mechanobiol. 2016;15(3):511-23.

10. Blanch-Mercader C, Guillamat P, Roux A, Kruse K. Integer topological defects of cell monolayers: Mechanics and flows. Phys Rev E. 2021;103(1-1):012405.

11. Guillamat P, Blanch-Mercader C, Kruse K, Roux A. Integer topological defects organize stresses driving tissue morphogenesis. bioRxiv. 2020:2020.06.02.129262.

12. Blanch-Mercader C, Guillamat P, Roux A, Kruse K. Quantifying Material Properties of Cell Monolayers by Analyzing Integer Topological Defects. Phys Rev Lett. 2021;126(2):028101.

13. Martino F, Perestrelo AR, Vinarský V, Pagliari S, Forte G. Cellular Mechanotransduction: From Tension to Function. Frontiers in Physiology. 2018;9(824).

14. Holle AW, Engler AJ. More than a feeling: discovering, understanding, and influencing mechanosensing pathways. Curr Opin Biotech. 2011;22(5):648-54.

15. Marullo S, Doly S, Saha K, Enslen H, Scott MGH, Coureuil M. Mechanical GPCR Activation by Traction Forces Exerted on Receptor N-Glycans. ACS Pharmacology and Translational Science. 2020;3(2):171-8.